# Genetic evidence for assortative mating on alcohol consumption in the UK Biobank

Laurence J. Howe[1,2]*, Daniel J. Lawson [1], Neil M. Davies [1], Beate St. Pourcain[1,3,4], Sarah J. Lewis [1], George Davey Smith [1,5] & Gibran Hemani [1,5]

Alcohol use is correlated within spouse-pairs, but it is difficult to disentangle effects of alcohol consumption on mate-selection from social factors or the shared spousal environment. We hypothesised that genetic variants related to alcohol consumption may, via their effect on alcohol behaviour, influence mate selection. Here, we find strong evidence that an individual's self-reported alcohol consumption and their genotype at rs1229984, a missense variant in *ADH1B*, are associated with their partner's self-reported alcohol use. Applying Mendelian randomization, we estimate that a unit increase in an individual's weekly alcohol consumption increases partner's alcohol consumption by 0.26 units (95% C.I. 0.15, 0.38; $P = 8.20 \times 10^{-6}$). Furthermore, we find evidence of spousal genotypic concordance for rs1229984, suggesting that spousal concordance for alcohol consumption existed prior to cohabitation. Although the SNP is strongly associated with ancestry, our results suggest some concordance independent of population stratification. Our findings suggest that alcohol behaviour directly influences mate selection.

[1] Medical Research Council Integrative Epidemiology Unit, Population Health Sciences, University of Bristol, Bristol, UK. [2] Institute of Cardiovascular Science, University College London, London, UK. [3] Max Planck Institute for Psycholinguistics, Nijmegen, The Netherlands. [4] Donders Institute for Brain, Cognition and Behaviour, Radboud University, Nijmegen, The Netherlands. [5] These authors contributed equally: George Davey Smith, Gibran Hemani. *email: laurence.howe@bristol.ac.uk

Human mate choice is highly non-random; spouse-pairs are generally more phenotypically similar than would be expected by chance[1–6]. Previous studies suggest that alcohol related phenotypes, ranging from consumption to alcohol dependence, are highly correlated within spouse-pairs[7–13]. However, the extent to which the spousal concordance is due to the effect of alcohol behaviour on mate selection (assortative mating) is currently unclear. Indeed, the spousal concordance may be related to assortment on other social and environmental factors (social homogamy) or be a consequence of an individual's partner influencing their alcohol behaviour after the individuals have paired up (partner interaction effects) or even relate to spousal similarities influencing relationship length (relationship dissolution)[11–13]. The mechanism explaining spousal concordance for alcohol consumption could have important implications relating to human social and reproductive behaviour. Figure 1 illustrates possible explanations for spousal concordance on alcohol consumption.

One biological mechanism that partially explains the phenotypic concordance between spouse-pairs is that they are on average more genetically similar across the genome than non-spouse-pairs[14]. Genotypes implicated in the aetiology of height, education, blood pressure and several chronic diseases have been shown to be correlated within spouse-pairs[15–18]. It is not known whether genetic variants implicated in alcohol metabolism, via their effect on alcohol behaviour, contribute to mate selection.

Alcohol behaviour has been shown to be highly heritable with estimates of 30–50% for alcohol use disorders[19,20] and a common variant heritability of 13% for self-reported alcohol consumption;[21] Genome-wide Association Studies (GWAS) have identified more than 15 loci implicated in either the aetiology of alcohol dependence[22–26] or alcohol consumption volume[21,24,27–29]. Notably, genetic variants in the Alcohol Dehydrogenase (ADH) and Aldehyde Dehydrogenase (ALDH) gene families are associated with differences in alcohol consumption[30]. For example, ADH1B is involved in the production of enzymes that oxidise alcohol and so individuals with certain alleles may find alcohol consumption unpleasant, resulting in lower intake. Similarly, a

genetic variant in ALDH2, rare in non-east Asian populations, is associated with a flush reaction to alcohol[31,32].

Alcohol consumption-related genetic variants can be useful to determine the most likely explanation for the spousal phenotypic concordance for alcohol use, by analogy with Mendelian randomization studies[33,34]. Genetic variants for alcohol consumption are in theory less susceptible to confounding from socio-economic and behavioural factors than measured alcohol consumption so can be used to rule out the possibility that social homogamy is driving the spousal phenotypic concordance[33,35]. The timing of the effects of alcohol consumption can be discerned by evaluating the spousal genotypic concordance for alcohol use-related variants. Genotypic concordance would imply that an effect exists prior to pairing, suggesting that some degree of the spousal phenotypic concordance is attributable to assortative mating (Fig. 2).

In this study we aim to explore spousal similarities for alcohol consumption using observational and genetic data. First, we estimate the association of an individual's self-reported alcohol use with the self-reported alcohol use of their partner. Second, we use a Mendelian randomization framework to estimate the effect of an individual's alcohol use on their spouse's alcohol use. Here, we use their partner's rs1229984 genotype, a missense mutation in ADH1B strongly associated with alcohol consumption as an instrumental variable for self-reported alcohol consumption. Third, we estimate the association of rs1229984 genotype between spouses, to evaluate the timing of possible causal effects, and investigate the possibility of bias from population stratification. Fourth, using the mean age of each couple as a proxy for relationship duration, we determine if there is an association between longer relationships and more similar spousal alcohol behaviour. As a positive control, to demonstrate the validity of derived spouse pairs and the usage of a Mendelian randomization framework, we also analyse height, known to be correlated between spouses, using similar methods.

We outline a framework using spouses and genetic data to evaluate assortative mating. We find that an individual's genotype at rs1229984 is associated with alcohol consumption in their

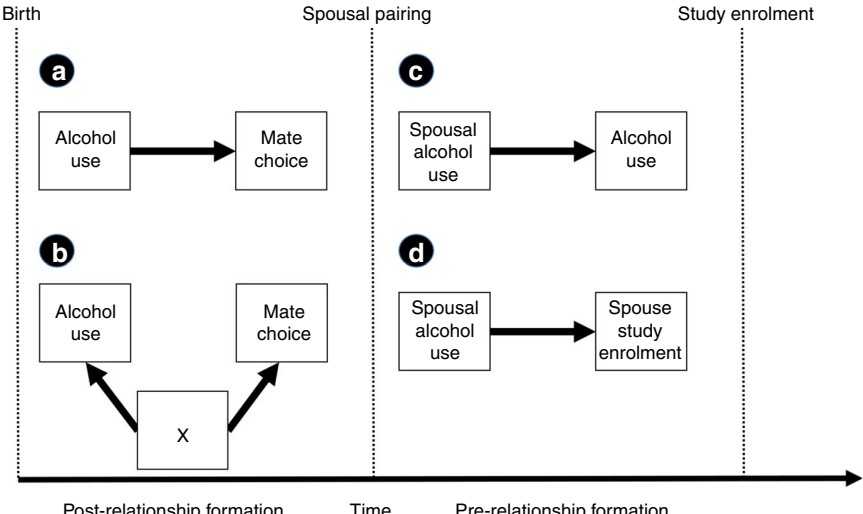

**Fig. 1** Possible explanations for spousal concordance on alcohol use. **a** Assortative mating. Alcohol behaviour influences mate choice; individuals are more likely to select a mate with similar alcohol consumption. **b** Social homogamy or confounding. An unknown confounder, X, influences mate-selection independent of alcohol behaviour. For example, ancestry or socio-economic status may influence both alcohol use and mate choice. **c** Partner interaction effects. During the relationship, spouses influence each other's alcohol consumption. For example, spousal alcohol consumption could become more similar over time. **d** Relationship dissolution. Spouse-pairs with more similar alcohol behaviour are more likely to remain in a relationship and be recruited into UK Biobank or similarly, are more likely to participate in the study together

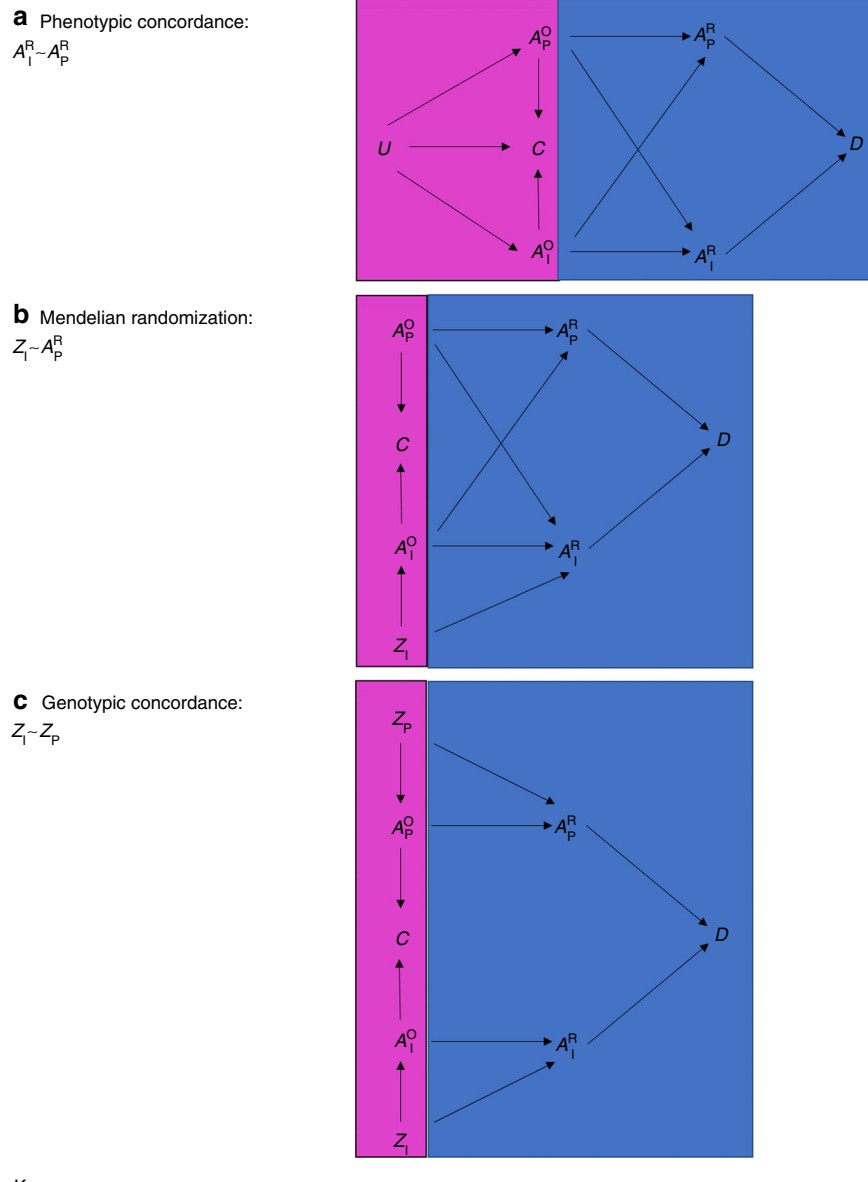

**Key**

$Z_{i,P}$: ADH1B genotype for an individual (I) and their partner (P).

$A_{I,P}^{O}$: alcohol consumption prior to partnering for an individual (I) and their partner (P).

$A_{I,P}^{R}$: alcohol consumption in the relationship for an individual (I) and their partner (P).

$C$ a measure of mate choice.

$D$ a measure of relationship duration.

$U$ representing unmeasured confounding factors.

Magenta: effects prior to relationship formation.

Blue: effects while spouses are in a relationship.

partner and their partner's rs1229984 genotype. We use several sensitivity analyses exploring the possibility of population stratification, to strengthen the case for assortative mating on alcohol consumption in the UK population.

## Results

**Spousal phenotypic concordance for height.** Measured height was strongly concordant between spouse-pairs. In a sample of 47,377 spouse-pairs, a 1 unit increase in an individual's height was associated with a 0.24-unit increase (95% C.I. 0.23, 0.25, $P < 10^{-16}$ for a linear relationship) in their partner's height. This

result is consistent with previous findings[36,37], validating the derived spouse pairs.

**Effect of height on height of partner.** The application of Mendelian randomization to spousal height was consistent with the previous evidence for assortative mating on height. Across 47,377 spouse-pairs, a 1 cm increase in an individual's height was associated with a 0.19 cm increase in their partner's height (95% C.I. 0.18, 0.21; $P < 10^{-16}$), distinctly smaller than the phenotype estimate (Z-test for difference of means: $P = 8.3 \times 10^{-8}$). The $I^2$ statistic (2.9%) and Cochran's Q test ($P = 0.64$) suggested consistent effects across SNPs, and estimates were consistent across

**Fig. 2** Disentangling mechanisms underlying spousal alcohol use similarities. **a** Phenotypic concordance. Spousal concordance for alcohol use during their relationship ($A_I^R$, $A_P^R$), as measured in UK Biobank, could be explained by several different possibilities. Assortative mating: Alcohol consumption prior to relationship formation ($A_I^O$, $A_P^O$) influences mate choice C. Comparing assorted pairs induces spousal correlations for $A^O$ and $A^R$. Partner interaction effects: Spouses may influence each other's alcohol behaviour over time while in a relationship. We represent this stochastic process by the arrows between alcohol use at relationship formation ($A_I^O$, $A_P^O$) and alcohol use at study entry ($A_I^R$, $A_P^R$). Note that effects likely relate to relationship length. Relationship dissolution: Spousal alcohol behaviour during the relationship ($A^R$ and $A_P^R$) influences the duration of the relationship D. Comparing non-dissolved pairs induces spousal correlations for $A^R$ in the remaining couples. Confounding factors: Unmeasured confounders U influence both C and $A^O$ leading to spousal correlation for $A^R$ independent of an effect of $A^O$ on C. **b** Mendelian randomization framework. An association between an individual's alcohol influencing genotype $Z_I$ and their spouse's alcohol use $A_P^R$ suggests that the spousal concordance is explained by assortative mating, partner interaction effects or relationship dissolution. Genetic variants are unlikely to be associated with socio-economic confounders suggesting that the confounding factors possibility is unlikely. **c** Genotypic concordance. Genotypic concordance for alcohol related genetic variants ($Z_I$, $Z_P$) suggests that some degree of the spousal concordance is explained by comparing assorted or non-dissolved pairs (assortative mating/ relationship dissolution). Partner interaction effects cannot lead to genotypic concordance because genotypes are fixed from birth

---

**Table 1 MR estimates for the effect of a 1 cm increase in height on partner's height**

| Test | Interpretation | Estimate (95% C.I.) | P-value |
|---|---|---|---|
| Phenotypic association for comparison | N/A | 0.24 (0.23, 0.25) | $<10^{-16}$ |
| Inverse-variance weighted | Primary causal estimate[a] | 0.19 (0.18, 0.21) | $<10^{-16}$ |
| Heterogeneity of Inverse-variance weighted | Balanced pleiotropy | $I^2 = 3.6\%$ | 0.68 |
| MR-Egger intercept | Intercept test for directional pleiotropy[b] | 0.001 (−0.006, 0.008) | 0.75 |
| MR-Egger regression | Regression estimate[a] | 0.19 (0.15, 0.21) | $<10^{-16}$ |
| Weighted median | Consistency[a] | 0.18 (0.15, 0.21) | $<10^{-16}$ |
| Weighted mode | Consistency[a] | 0.17 (−0.23, 0.57) | 0.41 |

[a]Units: mm change in partner's height per 1-unit increase in individual's height
[b]Units: Average pleiotropic effect of a height genetic variant on partner's height

---

the weighted median, weighted modal and MR-Egger estimators with the MR-Egger intercept test finding no strong evidence for directional pleiotropy (Table 1). Here, the interpretation of any directional pleiotropy would suggest that at least some of the SNPs associate with another phenotype either in the individual or the partner that in turn associates with the partner's height phenotype through a pathway other than the individual's height phenotype.

**Spousal genotypic concordance for height**. Similarly, the genotypic concordance analysis for height was strongly concordant with previous findings; we found strong evidence that spouses have similar genotypes at height influencing loci. Each 1 S.D. increase in an individual's height PGS was associated with a 0.024 S.D. higher PGS in their partner (95% C.I. 0.015, 0.033; $P = 1.96 \times 10^{-7}$ for a linear relationship).

**Phenotypic spousal concordance for self-reported alcohol use**. The majority of derived spouse-pairs had complete data for relevant self-reported alcohol behaviour phenotypes. Strong evidence was found for phenotypic concordance between spouse-pairs for all self-reported alcohol variables. Amongst 47,510 spouse-pairs, an individual self-reporting as a never-drinker was associated with higher odds (OR 13.03, 95% C.I., 10.98, 15.44 $P < 10^{-16}$ from logistic regression) of their partner self-reporting as a never-drinker. Similarly, when restricting to 42,844 pairs who both reported being current drinkers, an individual drinking three or more times a week had higher odds (OR 6.24, 95% C.I., 5.95, 6.54 $P < 10^{-16}$ from logistic regression) of their partner also drinking three or more times a week.

For self-reported alcohol consumption volume; 47,510 spouse-pairs had either complete phenotype data or reported their consumption frequency as less than weekly (in which case their weekly volume was assumed to be 0). After removing 189 pairs with outlying values (>5 S.D from the mean) from one or more members, the final sample included 47,321 spouse-pairs. In this

sample, each unit increase in an individual's weekly alcohol consumption volume was associated with a 0.37-unit increase (95% C.I. 0.36, 0.38 $P < 10^{-16}$ for a linear relationship) in the same variable in their partner.

**Effect of alcohol use on partner's alcohol use**. To evaluate the degree to which an individual's alcohol consumption is affected by their partner's genetically influenced alcohol consumption, we used a sample of 47,321 spouse-pairs with available data on weekly alcohol consumption. In this sample, each additional copy of the *ADH1B* major allele was associated with an increased weekly alcohol consumption of 3.99 units a week (95% C.I. 3.52, 4.45; $P < 10^{-16}$ for a linear relationship) in the same individual. Each additional copy of the major allele was associated with higher weekly alcohol consumption of 1.06 units a week (95% C.I. 0.59, 1.52; $P = 8.20 \times 10^{-6}$) in the reference individual's partner. After scaling the estimate using a Wald estimator; a 1 unit increase in an individual's alcohol consumption led to having partner's with alcohol consumption 0.26 units higher than baseline (95% C.I. 0.15, 0.38; $P = 8.20 \times 10^{-6}$). This effect is slightly lower than the phenotypic estimate of 0.37 units (95% C.I. 0.36, 0.38) although confidence intervals overlap (Z-test for difference of means: $P = 0.069$).

**Characteristics of rs1229984 in the UK Biobank**. In the sample of 385,287 individuals of recent European descent, the minor allele frequency (MAF) of rs1229984 was 2.8% and very strong evidence was found for the SNP violating Hardy-Weinberg Equilibrium (HWE) ($Chi^2 = 275$, $P < 10^{-16}$) due to fewer heterozygotes compared to expectation (expected = 20,972, observed = 20,194). However, when restricting to the sample of 337,114 individuals of British descent, the MAF of rs1229984 was 2.2% and there was little evidence of the SNP violating HWE ($Chi^2 = 2.0$, $P = 0.16$) and there were slightly more heterozygotes compared to expected (expected = 14,506 observed = 14,743) (Supplementary Table 1). Evidence was found of allele frequency

**Table 2 Meta-analysis of spousal-concordance for rs1229984 across centres**

| Recruitment centre | Number of spouse-pairs born within 100 km of each other | Beta (95% C.I.) |
| --- | --- | --- |
| Stockport | 9 | N/A[a] |
| Manchester | 662 | 0.024 (−0.088, 0.0675) |
| Oxford | 669 | −0.010 (−0.088, 0.067) |
| Cardiff | 930 | 0.022 (−0.043, 0.088) |
| Glasgow | 1046 | 0.072 (0.019, 0.125) |
| Edinburgh | 611 | −0.047 (−0.166, 0.070) |
| Stoke | 1215 | −0.012 (−0.075, 0.051) |
| Reading | 1352 | 0.003 (−0.055, 0.060) |
| Bury | 2244 | 0.012 (−0.031, 0.055) |
| Newcastle | 2976 | −0.025 (−0.064, 0.013) |
| Leeds | 2563 | 0.041 (0.001, 0.081) |
| Bristol | 2117 | 0.015 (−0.030, 0.060) |
| St Bartholomew's Hospital | 122 | −0.073 (−0.220, 0.074) |
| Nottingham | 2342 | 0.025 (−0.017, 0.066) |
| Sheffield | 2260 | 0.037 (−0.009, 0.082) |
| Liverpool | 2632 | 0.023 (−0.020, 0.066) |
| Middlesbrough | 1477 | 0.002 (−0.050, 0.053) |
| Hounslow | 838 | 0.073 (−0.000, 0.147) |
| Croydon | 1034 | 0.044 (−0.027, 0.115) |
| Birmingham | 1440 | −0.019 (−0.068, 0.031) |
| Swansea | 85 | −0.068 (−0.283, 0.146) |
| Wrexham | 29 | N/A[a] |
| Combined (Fixed effects) | 28,615 | 0.016 (0.004, 0.028) P = 0.011 |

[a]Linear regression estimates did not converge due to limited sample sizes, these studies were excluded from the meta-analysis

differences for rs1229984 between the two samples ($Chi^2 = 445$, $P < 10^{-16}$) suggesting that population substructure differences may explain the HWE results.

The SNP was found to be strongly associated with both genetic principal components and birth coordinates in both samples. In the less restrictive European sample, each additional major allele of rs1229984 was associated with being born 24.6 kilometres farther north (95% C.I. 22.2, 27.0) and 13.3 kilometres farther west (95% C.I. 12.1, 14.5). The SNP was similarly associated with principal components and birth coordinates in the sample of British descent although there were differences in effect estimates between the two samples (Supplementary Table 2). We also found strong evidence that self-reported alcohol consumption is strongly associated with birth coordinates and principal components in both samples concordant directionally with the SNP associations (Supplementary Table 3).

**Spousal genotypic concordance for alcohol use.** Amongst 47,549 spouse-pairs, strong concordance was observed for the genotype of rs1229984. Each additional copy of the major rs1229984 allele was associated with a higher number of major alleles in their partner (Beta 0.019; 95% C.I. 0.010, 0.028; $P = 5.0 \times 10^{-5}$ for linear relationship).

As a sensitivity analysis, we restricted the sample to 28,653 spouse-pairs born within 100 km of each other and stratified spouse-pairs by the 22 different UK Biobank recruitment centres. In this sample, we did not find strong evidence that birth location differences were associated with similarities in alcohol behaviour or rs1229984 genotype, contrasting with clear evidence of associations in the full spouse-sample. However, we did find evidence that genomic principal component differences were associated with spousal similarities for these variables, likely reflecting the fine-scale population structure of UK Biobank (Supplementary Table 4). Of the 22 centres, two centres were omitted from the meta-analysis because the limited sample sizes led to convergence issues in regression. A fixed-effects meta-

analysis was then used to estimate the spousal-concordance across the remaining 20 centres and 28,615 spouse-pairs. Evidence was found of spousal concordance for rs1229984 (Beta 0.016; 95% C.I. 0.004, 0.028; $P = 0.011$ for linear relationship), consistent with the previous analysis. Cochran's Q test for heterogeneity across the betas suggested no strong evidence for heterogeneity ($P = 0.34$) across the different centres (Table 2).

Furthermore, confidence intervals universally overlapped between Mendelian randomization and genotypic concordance estimates when comparing estimates from spouse-pair samples stratified on geographical birth proximity (born within 100 km and born more than 100 km apart) and the full spouse-pair sample (Supplementary Table 5).

**Relationship length and spousal alcohol use similarities.** We did not find strong evidence that higher mean couple age, used as a proxy for relationship length, was associated with more concordant spousal alcohol behaviour. Per 1-year increase in couple mean age, spousal differences in terms of weekly alcohol units consumed were 0.003 smaller (95% C.I. −0.021, 0.027, $P = 0.81$ for linear relationship). In terms of genotypic differences at rs1229984, we did not find strong evidence that older couples were more similar at the locus. Per 1-year increase in couple mean age, spousal allelic differences at rs1229984 were 0.00006 smaller (95% C.I. −0.00047, 0.00036; $P = 0.79$ for linear relationship).

**Discussion**

In this study, we used a large sample of derived spouse-pairs in a UK-based cohort to demonstrate that an individual's self-reported alcohol use and their genotype for an alcohol-implicated variant, rs1229984 in *ADH1B*, are associated with their partner's self-reported alcohol use. Furthermore, we showed that the genotype of the variant is concordant within spouse-pairs. There are several possible explanations for our findings. First, that rs1229984 influences alcohol behaviour, which has a

downstream effect on mate selection. Second, that a participant's alcohol use is influenced by their partner's alcohol use. Third, spouse-pairs with more similar alcohol behaviour were more likely to remain in a relationship, and so be present in our study sample. Fourth, that given the strong association of the SNP with both genetic principal components and birth coordinates, the spousal concordance is related to factors influencing social homogamy, independent of alcohol behaviour, such as place of birth, ancestry or socio-economic status. Indeed, the allele frequency of rs1229984 was found to deviate between European and British subsets of the UK Biobank.

However, we presented evidence suggesting that a substantial proportion of the spousal concordance is likely to be explained by the biological effects of the variant on alcohol consumption in the index individual. Firstly, we have tested the association between a causal SNP for alcohol consumption, and not the measured consumption itself, thereby avoiding any post-birth confounding factors suggesting that alcohol use has a direct effect on spousal alcohol use. Secondly, because rs1229984 is concordant between spouses, there must be some degree of assortment on alcohol consumption prior to cohabitation. Furthermore, we found little evidence to suggest that the mean age of each spouse-pair, used as a proxy for relationship length, was associated with alcohol behaviour similarities. These findings suggest that the spousal concordance is unlikely to be due to relationship dissolution after the age of 40. Thirdly, we accounted for possible effects of ancestral factors, which could have induced confounding, by including principal components as covariates in the Mendelian randomization analysis. Additionally, we conducted sensitivity analyses, including a within centre sensitivity analysis excluding spouse-pairs born more than 100 kilometres apart and a stratification of spouse-pairs by geographical birth proximity, finding consistent effect estimates.

The strong evidence for spousal-concordance on the variant has implications for conventional Mendelian randomization studies (i.e. estimating the causal effect of an exposure on an outcome)[33], which use the SNP as a genetic proxy for alcohol intake[38]. Assortative mating could lead to a violation of the Mendelian randomization assumption, that the genetic instrument for the exposure is not strongly associated with confounders of the exposure-outcome relationship. If both genetic and environmental factors affect alcohol consumption, then assortative mating on alcohol consumption could contribute to associations between genetic and environmental factors in the offspring, with the strength of association dependent on the degree of assortative mating[39].

Interestingly, the minor allele of rs1229984 (i.e. associated with lower alcohol consumption) has been previously found to be positively associated with years in education[38] and socio-economic related variables, such as the Townsend deprivation index and number of vehicles in household[40,41]. Each copy of the minor allele was associated with an additional 0.023 (95% C.I. 0.012–0.034) years of education and a 0.016 S.D. (95% C.I. −0.001 to 0.033) increase in intelligence[42,43]. These associations may be downstream causal effects of alcohol consumption, which implies that some of the spousal concordance for alcohol consumption could be explained by assortative mating on educational attainment[15] or alternatively these associations may reflect maternal genotype and intrauterine effects[44]. Over time, assortative mating on alcohol consumption may further strengthen the associations between rs1229984 and socio-economic related variables[39]. Of further interest is that the variant has previously been shown to be under selection[45] suggesting that the variant has historically had a substantial effect on reproductive fitness and may partially explain the violation of HWE observed across Europeans in our analyses.

The analyses in this study extended previous work on the concordance between spouse-pairs for alcohol behaviour[7–12] by comparing the phenotypic concordance with analyses utilising a genetic variant strongly associated with alcohol consumption. A major strength of this study is the use of distinct methods with different non-overlapping limitations, allowing for improved inference by triangulating the results from the different methods[46]. First, we evaluated the spousal phenotypic concordance for self-reported alcohol consumption, second we investigated the effect of an individual's rs1229984 genotype on the alcohol consumption of their spouse using Mendelian randomization, third we demonstrated spousal genotypic concordance for rs1229984 and fourth we explored whether older couples have more similar alcohol behaviour. The use of the UK Biobank dataset was a considerable strength for these analyses because of the low frequency of the rs1229984 minor allele; the large scale of the UK Biobank allowed for the identification of thousands of genotyped spouse-pairs. A further strength of these analyses is that we have demonstrated the utility of a Mendelian randomization framework for application to assortative mating by applying it to height and alcohol use. Indeed, the evidence for differences between the observational and Mendelian randomization estimates for spousal height suggest that the observational estimate may be inflated by confounding factors. A similar approach using polygenic scores has previously demonstrated assortative mating on educational attainment[18]. However, the use of Mendelian randomization has a notable advantage over polygenic approaches because of the possibility of using various sensitivity analyses to test for heterogeneity and consistency of the effect estimate[47–49].

There are several limitations of this study. First, although spouse-pairs were identified using similar methods to previous studies[15–17], the identified spouse-pairs have not been confirmed. However, the phenotypic spousal concordance estimate for height found in this study is highly concordant with previous estimates[36], consistent with derived couples being genuine. Second, despite follow-up analyses, it is difficult to definitively prove that the spousal concordance is a direct result of assortative mating on alcohol consumption. Assortment independent of alcohol use, potentially relating to ancestral or geographical factors, cannot be completely ruled out and downstream pleiotropic effects of the variant may influence mate selection. Third, the use of a single genetic instrument in the Mendelian randomization analysis, limited the use of sensitivity analyses[47–49] and meant it is not possible to infer similar associations for other alcohol-implicated variants. Fourth, selection into the UK Biobank, particularly with regards to participation of spouse-pairs is a potential source of bias[50]. Fifth, it is unclear whether the mean age of each couple is a suitable proxy for relationship length, which limits conclusions regarding the possibilities of partner interactions and relationship dissolution. Indeed, patterns of assortment on alcohol behaviour changing over time would confound the use of this proxy. Finally, it is difficult to extrapolate the results of this study in the UK Biobank to non-European populations. This is because of potential contextual influences; for example, in some East Asian populations, males are much more likely to consume alcohol than females[51,52]. Indeed, even within the UK, there may be regional variation that we were unable to detect in this study. Additionally, there is some evidence that the effect of genetic contributors to alcohol varies across different populations[29].

To conclude, our results suggest that there is non-random mating on rs1229984 in ADH1B, likely related to the effect of the variant on alcohol behaviour. These results suggest that alcohol use influences mate selection and argue for a more nuanced approach to considering social and cultural factors when examining causality in epidemiological studies. Further research investigating other alcohol-implicated variants, and other

societies and ethnicities, and assortment on other phenotypes, would strengthen these conclusions.

## Materials and methods

**UK Biobank**. UK Biobank is a large-scale cohort study, including 502,655 participants aged between 40–69 years. Study participants were recruited from 22 recruitment centres across the United Kingdom between 2006 and 2010[53,54]. For the purposes of our analyses, we restricted the dataset to a subset of 463,827 individuals of recent European descent with available genotype data, with individuals of non-European descent removed based on a k-means cluster analysis on the first four genetic principal components[55]. The different subsets of UK Biobank utilised in our analyses are illustrated in Supplementary Fig. 1.

The UK Biobank was approved by the North West Multi-centre Research Ethics Committee. All UK Biobank study participants gave informed consent. This research project was approved under an amendment to application 15825 and complied with all relevant ethical regulations.

**Spouse-pair subsample**. Spouse information is not explicitly available, therefore we used similar methods to previous studies[15–17] to identify spouse-pairs in the UK Biobank. Starting with the European subsample described above, household sharing information was used to extract pairs of individuals who (a) report living with their spouse (6141-0.0), (b) report the same length of time living in the house (699-0.0), (c) report the same number of occupants in the household (709-0.0), (d) report the same number of vehicles (728-0.0), (e) report the same accommodation type and rental status (670-0.0, 680-0.0), (f) have identical home coordinates (rounded to the nearest km) (20074-0.0, 20075-0.0), (g) are registered with the same UK Biobank recruitment centre (54-0.0) and (h) both have available genotype data. If more than two individuals shared identical information across all variables, these individuals were excluded from analysis. At this stage, we identified 52,471 potential spouse-pairs.

We excluded 4866 potential couples who were the same sex (9.3% of the sample), as unconfirmed same sex pairs may be more likely to be false positives. Although sexual orientation data were collected in UK Biobank, access is restricted for privacy/ethical reasons. To reduce the possibility that identified spouse-pairs are in fact related or non-related familial, non-spouse pairs; we removed three pairs reporting the same age of death for both parents (1807-0.0, 3526-0.0). Then we constructed a genetic relationship matrix (GRM) amongst derived pairs and removed 53 with estimated relatedness (IBD > 0.1). To construct the GRM; we used a pool of 78,341 markers, which were derived by LD pruning (50KB, steps of 5 KB, r2 < 0.1) 1,440,616 SNPs from the HapMap3 reference panel[56] using the 1000 Genomes CEU genotype data[57] as a reference panel. The final sample included 47,549 spouse-pairs.

**Non-spouse-pair samples**. For secondary analyses requiring data from unrelated individuals, we derived a sample of individuals of European descent and a more restrictive sample believed to be of white British descent. Starting with the UK Biobank subset of 463,827 individuals of recent European descent, we removed 78,540 related individuals (relevant methodology has been described previously[55]) to generate the European sample and using lists provided by UK Biobank, further restricted this sample to 337,114 individuals identifying as being of white British descent.

**Height and educational attainment**. At baseline, the height (cm) of UK Biobank participants was measured using a Seca 202 device at the assessment centre (ID: 50-0.0). Measured height was used as a positive control for the application of a Mendelian randomization framework in the context of assortative mating.

Educational attainment as characterised by years in full-time education was defined as in a previous publication[58]. Individuals born outside England, Scotland or Wales were removed because of schooling system differences, participants with a college or university degree were classified with a leaving age of 21 years and participants who self-reported leaving school when younger than 15 years were classified with a leaving age of 15. Educational attainment was included as a covariate in phenotypic analyses of spousal alcohol behaviour similarities as a possible confounder.

**Self-reported alcohol variables**. At baseline, study participants completed a questionnaire. Participants were asked to describe their current drinking status (never, previous, current, prefer not to say) (ID: 20117-0.0) and estimate their current alcohol intake frequency (daily or almost daily, three or four times a week, once or twice a week, one to three times a month, special occasions only, never, prefer not to say) (ID: 1558-0.0). Individuals reporting a current intake frequency of at least once or twice a week were asked to estimate their average weekly intake of a range of different alcoholic beverages (red wine, white wine, champagne, beer, cider, spirits, fortified wine) (ID: 1568-0.0, 1578-0.0, 1588-0.0, 1598-0.0, 1608-0.0).

From these variables, we derived three measures: ever or never consumed alcohol (current or former against never), a binary measure of current drinking for self-reported current drinkers (three or more times a week against less than three times a week) and an average intake of alcoholic units per week, derived by

combining the self-reported estimated intakes of the different alcoholic beverages consumptions across the five drink types, as in a previous study[21]. The questionnaire used the following measurement units for each of the five alcoholic drink types: measures for spirits, glasses for wines and pints for beer/cider, which were estimated to be equivalent to 1, 2 and 2.5 units, respectively. Individuals reporting current intake frequency of "one to three times a month", "special occasions only" or "never" (for whom this phenotype was not collected), were assumed to have a weekly alcohol consumption volume of 0. More information on alcohol variables used in this study is contained in Supplementary Table 6.

**Genotyping**. 488,377 UK Biobank study participants were assayed using two similar genotyping arrays, the UK BiLEVE Axiom™ Array by Affymetrix1 (N = 49,950) and the closely-related UK Biobank Axiom™ Array (N = 438,427). Directly genotyped variants were pre-phased using SHAPEIT3[59] and then imputed using Impute4 using the UK10K[60], Haplotype Reference Consortium[61] and 1000 Genomes Phase 3[57] reference panels. Post-imputation, data were available on ~96 million genetic variants.

**Utilising genetic data to disentangle spousal correlations**. In general, the effects of genetic variation on a phenotype can be assumed to be via the variant's effect on intermediary observable or unobservable phenotypes. In the context of assortative mating, it is unlikely that individuals would assort based directly on genotype but rather on an observed phenotype influenced by genetic factors. Assuming that a phenotype is influenced by genetic factors G and individuals assort on the phenotype such that the phenotypic correlation between spouses is equal to C, then expected correlations between an index individual's G and their partner's phenotype and G induced by assortment can be shown to be a function of the heritability of the phenotype and the spousal phenotypic correlation C (Supplementary Methods). This implies that estimates of assortative mating utilising genetic data are likely to be attenuated compared to the true value of phenotypic assortment, unless genetic factors completely explain variation in the phenotype of interest or the estimates are rescaled as in Mendelian randomization.

However, there are notable advantages of applying genetic approaches such as Mendelian randomization and genetic correlation analyses to the context of assortative mating for mechanistic understanding. In conventional Mendelian randomization studies[33,34], genetic variants are used as proxies for a measured exposure to evaluate potential causal relationships between an exposure and an outcome (e.g. LDL cholesterol and coronary heart disease[38]). Genetic proxies may be more reliable than the measured exposure because of the reduced potential for confounding and reverse causation.

In the context of Mendelian randomization across spouses, the premise is largely similar; the exposure is an individual's phenotype (e.g. alcohol consumption), proxied by a genetic instrument, and the outcome is their partner's phenotype (e.g. alcohol consumption). A Mendelian randomization approach can evaluate a direct effect of an individual's alcohol consumption on the alcohol consumption of their partner as opposed to effects of social homogamy. A direct effect captured by a Mendelian randomization framework could capture; individuals being likely to select a mate with similar behaviour (assortative mating); an individual's alcohol consumption influencing their partner's during the relationship (partner interaction effects) or more similar couples staying together for longer (relationship dissolution). Note that as genotype is fixed from birth, a Mendelian randomization estimate will not capture an effect of the partner's alcohol consumption on the index individual during the relationship. Interpretation can be nuanced, as for example, it seems unlikely that an individual's height could influence the height of their partner, but partner interaction effects are highly plausible for alcohol behaviour.

Similarly, estimating the genotypic concordance between spouses for variants relating to a trait of interest can be used to improve mechanistic understanding. The interpretation of genotypic concordance is comparable to that of Mendelian randomization across spouses with two important distinctions. First, genotypic concordance will not capture partner interaction effects as germline DNA is fixed for both spouses prior to assortment. Second, concordance induced by assortment will be further attenuated compared to a Mendelian randomization approach.

**Spousal phenotypic spousal concordance for height**. To verify the validity of the derived spouse-pair sample, we evaluated the spousal phenotypic concordance for height. Previous studies have found strong evidence of spousal concordance for height, so comparable results would be consistent with derived spouses being genuine. The spousal phenotypic concordance was estimated using a linear regression of an individual's height against the height of their partner, adjusting for sex. With one unique phenotype pairing within couples (male spouse height/female spouse height), each individual in the dataset was included only once as either the reference individual or their partner.

**Effect of height on height of partner**. We validated the application of a Mendelian randomization approach to assortative mating using height as a positive control; genotypes influencing height have previously demonstrated to be highly correlated between spouse-pairs[15]. As a measure of genetically influenced height, we started with 382 independent SNPs, generated using LD clumping (r2 < 0.001)

in MR-Base[62], from a recent Genome-wide Association Study (GWAS) of adult height in Europeans[63].

For the purposes of the Mendelian randomization analysis, we restricted analyses to spouse-pairs with complete measured height data and genotype data. First, we estimated the association between 378 SNPs (four SNPs were unavailable in the QC version of the dataset) and height in the same individual, using the spouse-pair sample with sex included as a covariate. Second, we estimated the association between the 378 SNPs and spousal height. PLINK[64] was used to estimate the SNP-phenotype associations also including sex as a covariate. We then estimated the effect of a 1 cm increase in an individual's height on their partner's height using the TwoSampleMR R package[62] and the internally derived weights described above. The fixed-effects Inverse-Variance Weighted (IVW) method was used as the primary analysis. Cochran's Q test and the $I^2$ statistic were used to test for heterogeneity in the fixed-effects IVW[65]. MR Egger[47] was used to test for directional pleiotropy. The weighted median[48] and mode[49] were used to test the consistency of the effect estimate. With two unique pairings between genotype and phenotype in each couple (male spouse genotype/ female spouse height and the converse), each individual in the dataset was included twice as both the reference individual and as the partner.

**Spousal genetic concordance for height**. To evaluate spousal genotypic concordance for height, we evaluated the association between height polygenic scores (PGS) across spouse-pairs. Height PGS were constructed in PLINK[64] using the 378 height SNPs discussed above. The cross-spouse association was estimated using linear regression of an individual's PGS against the PGS of their partner. With one unique genotype pairing within couples (male spouse genotype/female spouse genotype), each individual in the dataset was included only once as either the reference individual or their partner.

**Phenotypic spousal concordance for self-reported alcohol use**. To evaluate the phenotypic concordance on alcohol use we compared self-reported alcohol behaviour between spouses. We estimated the spousal concordance for the two binary measures (ever or never consumed alcohol, three or more times a week) using a logistic regression of the relevant variable for an individual against the relevant variable for their partner, adjusting for sex, age and partner's age. In addition, we included recruitment centre, height and education (of both spouses) in the model as potential confounders. Similarly, linear regression was used to estimate the spousal-concordance for continuous weekly alcohol consumption volume, adjusting for the same covariates. Spouse-pairs with any missing phenotype data, or where one or more spouses reported their weekly alcohol consumption volume to be more than five standard deviations away from the mean (calculated using the sample of individuals with non-zero weekly drinking) were removed from relevant analyses. With one unique phenotype pairing within couples (male alcohol variable/ female alcohol variable), each individual in the dataset was included only once as either the reference individual or their partner.

**Effect of alcohol use on partner's alcohol use**. We then applied the Mendelian randomization framework to investigate if an individual's genotype at rs1229984 in ADH1B affects the self-reported alcohol consumption volume of their partner. Given the rarity of individuals homozygous for the minor allele in European populations, the MAF is 2.9% in the 1000 Genomes CEU population[57], we first determined whether an additive or a dominant model (as used in previous studies[38,66]) was most appropriate for the SNP by comparing the association of genotype at rs1229984 with self-reported weekly alcohol consumption in the European and British samples. We found strong evidence to suggest that the SNP has an additive effect on alcohol consumption (Supplementary Table 7) and assumed this model in all relevant analyses.

For the Mendelian randomization analysis, we restricted analysis to spouse-pairs where both members had genotype data, and one or more members had self-reported alcohol consumption volume. First, we estimated the association of the rs1229984 genotype with alcohol consumption in the same individual after adjusting for sex, age, centre and the first 10 principal components of the reference individual. Second, we estimated the association between rs1229984 and spousal alcohol consumption after adjusting for sex, age (of both spouses), centre and the first 10 principal components of both spouses. PLINK[64] was used to estimate the SNP-phenotype associations. We then estimated the effect of a 1 unit increase in an individual's weekly alcohol consumption volume on the same variable in their partner. The Wald ratio estimate was obtained using mr_wald_ratio function in the TwoSample MR R package[62] using internally derived weights. Sensitivity analyses were limited due to the use of a single genetic instrument. With two unique pairings between genotype and phenotype in each couple (male alcohol variable/ female genotype and the converse), each individual in the dataset was included twice as both the reference individual and as the partner.

**Spousal genotypic concordance for alcohol use**. We then investigated properties of the rs122984 variant in the UK Biobank that may be relevant to assortative mating. Starting with the UK Biobank subset of 463,827 individuals of recent European descent, we removed 78,540 related individuals, which were identified using an algorithm applied to the related pair list provided by UK Biobank (third

degree or closer)[55], and tested Hardy-Weinberg Equilibrium (HWE) in the resulting sample of 385,287 individuals. To evaluate the possibility of population stratification, we investigated the association of both the SNP and self-reported alcohol consumption with genetic principal components and birth coordinates. As a sensitivity analysis, we also restricted the sample to a more homogeneous sample of British individuals, provided by the UK Biobank, and repeated analyses.

We then estimated the genotypic concordance between derived spouse-pairs for rs1229984 genotype using linear regression. As a sensitivity analysis, we then investigated the possibility that spousal-concordance for rs1229984 was driven by fine-scale assortative mating due to geography, which is itself associated with genetic variation within the UK[67,68]. For this, we restricted the sample to include only 28,653 spouse-pairs born within 100 km of each other. To test the validity of this sensitivity analysis, we explored whether birth or genetic differences (as determined by principal components) between spouses are associated with alcohol behaviour or rs122984 genotype differences in the restricted and full spouse-pair samples. The spouse-pairs were then stratified into the 22 different UK Biobank recruitment centres and logistic regression analyses were re-run to estimate the spousal-concordance of the ADH1B genotype by centre. With one unique genotype pairing within couples (male genotype/female genotype), each individual in the dataset was included only once as either the reference individual or their partner. Geographical patterns of heterogeneity across the different UK Biobank recruitment centres would provide evidence of population stratification.

As a further sensitivity analysis to explore potential population stratification bias, we compared Mendelian randomization and genotypic concordance estimates between the sample of 28,653 spouse-pairs born within 100 km of each other with estimates from the sample of 13,770 pairs born more than 100 kilometres apart, and with the full sample of 47,549 spouse-pairs. Note a subset of spouse-pairs did not have complete birth coordinate data.

**Relationship length and spousal alcohol use similarities**. Relationship length may influence spousal similarities for alcohol behaviour because spouses become more similar over time or because pairs with similar alcohol behaviour tend to have longer relationships. To explore these possibilities, we investigated the association between relationship length and alcohol behaviour and rs1229984 genotype similarities. Without available data on relationship length, we used the mean age of each couple as a proxy and evaluated associations using a linear regression of mean couple age against spousal difference in weekly alcohol consumption and rs1229984 genotype. Analyses were adjusted for the sex of reference individual.

## Data availability
This study used data from the UK Biobank, a list of derived spouse-pairs has been returned to the study. For details please contact access@ukbiobank.ac.uk. All other data are contained in the article and its supplementary information or available upon reasonable request.

## Code availability
Code for deriving the spouse-pair sample in UK Biobank is available at (github.com/LaurenceHowe/SpousePairs). Additional code is available at request (Laurence.Howe@bristol.ac.uk).

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

## Acknowledgements

L.J.H. was a Medical Research Council funded PhD student at the University of Bristol and is now funded by the British Heart Foundation and University College London. N.M.D., S.J.L. and G.D.S. work in the Medical Research Council Integrative Epidemiology Unit at the University of Bristol (MC_UU_00011/1), which is supported by the Medical Research Council and the University of Bristol. N.M.D. is supported by the Economics and Social Research Council (ESRC) via a Future Research Leaders grant [ES/N000757/1]. D.J.L. [WT104125MA] and G.H. [208806/Z/17/|] are both supported by the Wellcome Trust. B.T.S.P. is supported by the Max Planck Society and the Simons Foundation (Award ID: 514787). UK Biobank received ethical approval from the Research Ethics Committee (11/NW/0383). This research was approved as part of application 15825 (PI: Dr Philip Haycock).

## Author contributions

L.J.H., G.D.S., G.H. and D.J.L. formulated the project outline and analysis plan. L.J.H. performed all statistical analyses and drafted the first manuscript draft under supervision from G.D.S., G.H., S.J.L., B.S.P. and N.M.D. All authors contributed to interpretation of results and writing of the final manuscript.

## Competing interests

N.M.D. reports a grant for research unrelated to this work from the Global Research Awards for Nicotine Dependence (GRAND), an independent grant making body funded by Pfizer. The remaining authors declare no competing interests.
