## [Peer Review File · Nature Communications]

Reviewers' Comments:

Reviewer #1:

Remarks to the Author:

The paper focuses on alcohol consumption and mate choice, and aims to disentangle the nature of the relation between the two. The authors hypothesize that genetic variants related to alcohol consumption affect mate selection, and do so through alcohol behavior.

The study uses a sample of 47,000 spouse-pairs in the UK Biobank, and involves state-of-the-art genetic analyses. Noteworthy is the application of Mendelian Randomization (MR) to examine effects across spouses.

Findings include evidence for cross-spousal associations for both self-reported alcohol use and the rs1229984 genotype. The genotype association within spouse-pairs suggests that some spousal similarity was present prior to cohabitation, and supports a model of alcohol behavior influencing mate selection.

In general, I believe this is a well-written paper, with sound methodology, original contributions and important findings. I do however have some concerns about the current manuscript.

1. The nature of the relation between alcohol use and mate choice

The authors refer to previous findings of phenotypic alcohol use being correlated between spouses, and outline three major mechanisms and explanations for this association: a) social homogamy, b) partner interaction effects and c) assortative mating. The findings are interpreted as support for assortative mating – genetic similarity in disposition to alcohol use contributes to mate selection. The interpretation and conclusion makes a lot of sense, but I wonder if other mechanisms also could explain the finding.

Given the age of the sample (40-69 years old) it is reasonable to assume that most of the identified couples have in fact been spouses for some time. Others, who initially became couples, but subsequently broke up, are not included in the sample. Thus, being defined as a couple requires both establishing a relationship in the first place, and then choosing to remain spouses - for years, or even decades. The implication could be that genetic similarity (in rs1229984) does not necessarily explain the phenomenon of becoming a couple, but rather the phenomenon of remaining a couple. If there was in fact random mating (for rs1229984) but non-random relationship dissolution (i.e. dissimilarity in rs1229984 contributing to break-ups), we would also expect to see the spousal similarity among established couples as observed in the study.

I believe the paper would be strengthened by addressing this issue, preferably by including some new analyses involving time or age (or other relevant variables), or at least by discussing the process of relationship dissolution and continuation.

2. Does alcohol behavior directly influence mate selection?

A major conclusion of the study appears to be that alcohol behavior directly influences mate selection (line 43-44, 449). I believe it would be helpful with some more elaboration of this conclusion, and also consideration of alternatives. Following the point made in #1, it might be argued that three different phenotypes are studied here: a) initial choice of mate, b) continuing a relationship, and c) drinking behaviors.

Whereas the analyses of genotype correlations pertain to a) and/or b), the MR analyses appear to address the question of whether phenotypic drinking in one person influences spousal drinking. The use of MR to examine the nature of associations across spouses is rather novel, and interesting, but unless I misunderstand the analyses, the aim is to determine if the exposure variable (spouse1 drinking) is causally related to the outcome (spouse2 drinking). The findings suggest indeed a causal relationship, assumedly reciprocal. It is however not fully clear how this phenotypic causality, supported by the MR, also supports the main conclusion of alcohol behavior influencing mate selection. I believe the paper would benefit from a bit more theoretical outline and also info about the logic of MR across spouses, and some more elaboration of the convergence or divergence of the findings from MR vs the genotype correlation analyses.

3. Possibility of horizontal pleiotropy

The rs1229984 genotype has been shown to be associated with alcohol use, previously and in the current study. But what if rs1229984 is also associated with for example drug use or nicotine dependence? Any of these, or other associated phenotypes, could in theory be a driving force in the mate choice (or mate retaining) process. Admittedly, the authors states in the limitations (line 437-439) that 'Assortment independent of alcohol use cannot be completely ruled out...'. I believe, however, the paper would benefit from some more attention to this issue.

If the effect of rs1229984-similarity is about alcohol use we would expect to see no remaining rs1229984-correlation across spouses, after controlling for phenotypic alcohol behavior. In contrast, if there is still a spousal rs1229984-correlation after controlling for alcohol behavior, there would be reason to look for other associated phenotypes that mediate the effect. I believe such analyses could easily be performed and would add to the validity of findings.

4. Minor issues

- i) It would be helpful with some more descriptive statistics, e.g. the distribution of the drinking variables.
- ii) The analyzed sample consists of (assumed) spouses, but same-sex spouses are excluded. It would be helpful to provide arguments for this exclusion.
- iii) Alcohol behavior is reported to be highly heritable (line 69) but no references are included. It would be good to cite e.g. some twin studies and to include heritability estimates.
- iv) Analyzing height as a way of validating the subsequent alcohol analyses represents a strength. However, it is not fully clear why height is subjected to phenotypic correlation analyses and MR, but not genotypic correlations across spouses.
- v) The authors seem to use the term 'correlation' when referring to an association (e.g. "We estimated the spousal correlation ... using a logistic regression", line 179-181). This might be fine, but many readers would expect a correlation to mean e.g. 'Pearson correlation' or 'polychoric correlation'. It might be helpful to a) either refer to associations as 'associations' rather than 'correlations', or b) use 'correlations' but then also include estimates of Pearson and tetrachoric correlations (which would make the findings even more comparable to previous studies reporting such correlations).

Reviewer #2:

Remarks to the Author:

The paper by Howe et al is an interesting example of the application of Mendelian Randomization to understand the correlation between spouses with regard to alcohol consumption. They conclude that there is some correlation and that some of this is due to genetics - that assortative mating matches people with similar consumption of alcohol. I think that these questions probably have already a weight of observational evidence behind them - the novelty of this paper is the application of genetic methods to this question, this is in an attempt to disentangle whether this correlation existed before or after the people became spouses.

I notice that they initially frame the paper with a public health angle, including in the first paragraph of the introduction. I think that this is possibly less relevant - I don't think that the finding that spouses have similar health-related behaviors is particularly novel, and the public health relevance is not strengthened by demonstrating this with genetic instruments.

One thing that might be of public health interest is the balance of the direction of the effect. If high consuming partners increase consumption in their spouses or if tee-total people decrease the effect in theirs. One way to look at this might be to specifically consider the patterns of alcohol consumption within couples discordant for the variant studied here. This might suggest that you get a double benefit from treating one member of a couple, or that the people might slip back into old habits if both aren't considered.

In terms of the analysis presented, there are a few things I would suggest. Firstly there seems to

me that there is an risk in this analysis of the result being driven by assortative mating other than that based on alcohol consumption, a point that the authors acknowledge in the discussion. This is also suggested by the fact that the variant in question was associated with both some of the principle components and with birth site. What does the association look like when the values for one or both partners for these principle components are included in the model?

For the analysis presented in Table 2, I notice that there seems to be a large difference in the numbers of spouse pairs included in this analysis from different regions. I suspect that this reflects different rates of migration to different cities (resulting in couples from nearer or further apparent), to what extent do the authors feel that this suggests limitations to the generalisability of the results? This locus is known to be ancestry specific, as are the social factors that lead to assortative mating. And, this may even be different in different parts of the UK. While there isn't any evidence of heterogeneity in the analysis, this might simply be due to a lack of diversity in the samples available in this analysis.

I'm curious to know if UK Biobank had a policy of people bringing their spouses along - was there any ad-hoc enrollment of this sort? I'm not sure that this would cause an issue with the analysis, but I can see that it might result in a sample were spouses were more similar on a range of factors than the general population.

The statistical analysis excluded those more than 5SDs. Why was this done? Do the authors think that this data is less reliable? If there interest was in public health, these potentially alcohol abusers might be a particular group to target.

Reviewer #3:

Remarks to the Author:

Howe et al. present an analysis of correlations between couple's alcohol consumption using observational and genetic data from the latest release of the UK Biobank. In particular, they compare how the alcohol consumption is correlated between couples from observations, and using a SNP associated with alcohol consumption as instrumental variable. They also check how this SNP is correlated between couples. They repeat some of these approaches for height. The authors present interesting results. In addition to alcohol consumption behaviour, the results may help to understand the complex structure in cohorts such as UK Biobank and how to deal with it. I also found interesting the Discussion section. One of my major concerns is related to the correlation between the SNP genotypes used as instrumental variable and the first principal components or birth coordinates.

1) The authors observe that the SNP rs1229984 is significantly associated with the first Principal Components (PC), and birth coordinates (BC). As the authors already comment, this may be due to a population structure (which may be both local and/or global). Furthermore, this SNP may be pleiotropic. This could considerably affect the interpretation of the results. The authors address this point by performing analyses grouping individuals by assessment center. However, I think stronger evidence is needed to study how this stratification may affect the results. Below I make some suggestions which, in my opinion, would help to address this problem:

1.a) The authors showed that the rs1229984 genotypes are correlated with PC/BC. I think it would be useful to know if alcohol consumption is also associated with PC/BC.

1.b) I think that additional variables have to be included in the models that estimate the SNP effect. In addition to sex, the first principal components, age (I think that age, and partner age may also be relevant when computing correlation of alcohol consumption between partners), genotyping array and assessment center should also be included. In addition, given the stratification of the SNP, and that there are relatives in the analysis, I think that the ordinary linear

models (i.e. the non-logistic models used for estimating the SNP effect), would benefit from using Mixed Linear Models to correct by the Genetic Relationship Matrix (using a schema such as leave-one-chromosome-out, LOCO, where the chromosome of the tested SNP is excluded from the GRM, otherwise they may lose power. Tools such as BOLT-LMM will do this automatically).

1.c) Given that rs1229984 is stratified, why the authors did not also used other alcohol consumption associated SNPs in the same way the authors do with different SNPs associated with height?

1.d) The authors address the problem of the SNP stratification by performing analyses by assessment center. I think these analyses provide some support for the observed results not being artifacts from stratification. However, I think it would also be good to show that the genotype of the individuals grouped by assessment center, are not associated with their PC/BC. In addition, I would use a somewhat inverse approach. Are the genotypes of the couples that were born 100 or 200 (a number as larger as possible, but that still retains a considerable sample size) miles away still correlated? If so, is their alcohol consumption also correlated? If the answers to this question is yes, then: what is the effect of rs1229984 on the partner alcohol consumption for this couple?

2) Related to the model used for estimating the effect of the SNP rs1229984 on alcohol consumption, the rationale behind using a dominant model vs an additive model is not clear to me. Although the MAF is small, I think that with this sample size, there will be enough individuals with two minor alleles to fit the models. Although, there will be a small number of individuals (~100?) with two minor alleles, the effect of the minor/major allele may be under/over estimated. Maybe it is not relevant, but without checking, it is hard to test their potential effect. Given that the SNP effects estimated in the model may be relevant for the Mendelian Randomization approach, and given that some confidence intervals are overlapping by a small margin, i think it would be relevant to at least test whether the estimated SNP effects on alcohol consumption are affected by fitting an additive model.

Minor points:

- In figure 1, would it make sense to add an additional possibility after spousal paring? A diagram were a third factor "X" affecting alcohol consumption of both partners (a factor which may also be correlated with genetic variants)?.

- In figure 1, the legend is labeled A), B), and C), but the figure as 1), 2), and 3).

- The authors comment about rs1229984 violating the HWE hypothesis. As a suggestion, I would also add that HWE may be violated under selection (the authors point out that this SNP may be under selection in the Discussion), In addition, I think the HWE threshold should be much less restrictive when the sample sizes start to be large enough (and the population structure potentially increases its effect on the test). For instance, on the PLINK2.0 website they comment about this (and they suggest to use a small threshold for filtering, <https://www.cog-genomics.org/plink/2.0/filter#hwe>).

- The authors use several variables to decide whether two individuals are part of a couple. However, they do not use shared household information which may be available under request. If it is not too much hassle, it would be interesting to see whether the computed couples agree with the shared household information.

- It would be easier for the reader if the author indicate the UK Biobank codes of the used fields,

when these are first described in the methods.

- I think there is a typo in the line 192: an -> a.

- In line 312, I believe it would be clearer if it says "...than the reference group.", "...the partners of the reference group."

- Overall the manuscript is clearly written and easy to understand. However, I had sometimes the feeling that the manuscript is a bit disordered. Maybe putting the sections about height in the Results section together would help. If the Methods will be in the main text, there are some bits that also seem a bit redundant. However, this is just my opinion, which may be based on a personal preference, so feel free to ignore it if the authors disagree.

Reviewer 1

Comment 1.1

1. The nature of the relation between alcohol use and mate choice

The authors refer to previous findings of phenotypic alcohol use being correlated between spouses, and outline three major mechanisms and explanations for this association: a) social homogamy, b) partner interaction effects and c) assortative mating. The findings are interpreted as support for assortative mating – genetic similarity in disposition to alcohol use contributes to mate selection. The interpretation and conclusion makes a lot of sense, but I wonder if other mechanisms also could explain the finding.

Given the age of the sample (40-69 years old) it is reasonable to assume that most of the identified couples have in fact been spouses for some time. Others, who initially became couples, but subsequently broke up, are not included in the sample. Thus, being defined as a couple requires both establishing a relationship in the first place, and then choosing to remain spouses - for years, or even decades. The implication could be that genetic similarity (in rs1229984) does not necessarily explain the phenomenon of becoming a couple, but rather the phenomenon of remaining a couple. If there was in fact random mating (for rs1229984) but non-random relationship dissolution (i.e. dissimilarity in rs1229984 contributing to break-ups), we would also expect to see the spousal similarity among established couples as observed in the study.

I believe the paper would be strengthened by addressing this issue, preferably by including some new analyses involving time or age (or other relevant variables), or at least by discussing the process of relationship dissolution and continuation.

Response

We agree, spousal similarities for alcohol behaviour could plausibly influence relationship dissolution and continuation. Intuitively, couples that differ greatly in alcohol consumption may be less likely to remain in a relationship long enough to be recruited into UK Biobank together. To explore this possibility, we have added extra analyses as suggested.

Using the mean age of each couple as a proxy for relationship duration, we explored whether relationship length is positively associated with spousal alcohol behaviour and rs1229984 genotype similarities. We did not find strong evidence that older couples were more phenotypically or genotypically similar; contrastingly finding some evidence that older couples are actually more genotypically dissimilar at the locus of interest. Stronger conclusions are limited because of the use of mean couple age as a proxy for relationship length; a notable issue is that if assortative mating patterns on alcohol have changed over time, this would induce an association between mean age and spousal similarities unrelated to relationship dissolution. Nevertheless, these analyses are inconsistent with relationship dissolution (and indeed partner interaction effects) explaining a substantial proportion of the observed spousal phenotypic concordance. We have added relevant sections to the manuscript pertaining to these analyses, with a few examples included below.

Results

“We did not find strong evidence that increased mean couple age, used as a proxy for relationship length, was associated with more concordant spousal alcohol behaviour. Per 1-year increase in couple mean age, spousal differences in terms of weekly alcohol units consumed were 0.017 smaller (95% C.I. -0.040, 0.007, $P=0.16$). In terms of genotypic differences at rs1229984, we found weak evidence that older couples are more dissimilar at the locus. Per 1-year increase in couple mean age, spousal allelic differences at rs1229984 were 0.0004 larger (95% C.I. 0.0000, 0.0009; $P=0.035$).”

Discussion

“Furthermore, we found little evidence to suggest that the mean age of each spouse-pair, used as a proxy for relationship length, was associated with alcohol behaviour similarities. These findings suggest that the spousal concordance is unlikely to be due to relationship dissolution after the age of 40.”

Comment 1.2

2. Does alcohol behavior directly influence mate selection?

A major conclusion of the study appears to be that alcohol behavior directly influences mate selection (line 43-44, 449). I believe it would be helpful with some more elaboration of this conclusion, and also consideration of alternatives. Following the point made in #1, it might be argued that three different phenotypes are studied here: a) initial choice of mate, b) continuing a relationship, and c) drinking behaviors.

Whereas the analyses of genotype correlations pertain to a) and/or b), the MR analyses appear to address the question of whether phenotypic drinking in one person influences spousal drinking. The use of MR to examine the nature of associations across spouses is rather novel, and interesting, but unless I misunderstand the analyses, the aim is to determine if the exposure variable (spouse1 drinking) is causally related to the outcome (spouse2 drinking). The findings suggest indeed a causal relationship, assumedly reciprocal. It is however not fully clear how this phenotypic causality, supported by the MR, also supports the main conclusion of alcohol behavior influencing mate selection. I believe the paper would benefit from a bit more theoretical outline and also info about the logic of MR across spouses, and some more elaboration of the convergence or divergence of the findings from MR vs the genotype correlation analyses.

Response

We have added sections to the manuscript detailing the rationale behind Mendelian randomization in this context and discussing the differences in interpretation between genotypic concordance and Mendelian randomization. We also used mathematical equations and simulations (contained in the Supplementary Methods) to demonstrate that phenotypic assortment induces spousal genotypic associations dependent on the heritability of the trait and degree of assortment. Figure 2 provides the theoretical basis for our analyses.

Methods

“In general, the effects of genetic variation on a phenotype can be assumed to be via the variant’s effect on intermediary observable or unobservable phenotypes. In the context of assortative mating, it is unlikely that individuals would assort based directly on genotype but rather on an observed phenotype influenced by genetic factors. Assuming that a phenotype is influenced by genetic factors G and individuals assort on the phenotype such that the phenotypic correlation between spouses is equal to C , then expected correlations between an index individual’s G and their partner’s phenotype and G induced by assortment can be shown to be a function of the heritability of the phenotype and the spousal phenotypic correlation C (**Supplementary Methods**). This implies that estimates of assortative mating utilising genetic data are likely to be attenuated compared to the true value of phenotypic assortment, unless genetic factors completely explain variation in the phenotype of interest.

However, there are notable advantages of applying genetic approaches such as Mendelian randomization and genetic correlation analyses to the context of assortative mating for mechanistic understanding. In conventional Mendelian randomization studies³³³⁴, genetic variants are used as proxies for a measured exposure to evaluate potential causal relationships between an exposure and an outcome (e.g. LDL cholesterol and coronary heart

disease⁴⁵). Genetic proxies may be more reliable than the measured exposure because of the reduced potential for confounding and reverse causation.

In the context of Mendelian randomization across spouses, the premise is largely similar; the exposure is an individual's phenotype (e.g. alcohol consumption), proxied by a genetic instrument, and the outcome is their partner's phenotype (e.g. alcohol consumption). A Mendelian randomization approach can evaluate a direct effect of an individual's alcohol consumption on the alcohol consumption of their partner as opposed to effects of social homogamy. A direct effect captured by a Mendelian randomization framework could capture; individuals being likely to select a mate with similar behaviour (assortative mating), an individual's alcohol consumption influencing their partner's during the relationship (partner interaction effects) or more similar couples staying together for longer (relationship dissolution). Interpretation can be nuanced, as for example, it seems unlikely that an individual's height could influence the height of their partner, but partner interaction effects are highly plausible for alcohol behaviour.

Similarly, estimating the genotypic concordance between-spouses for variants relating to a trait of interest can be used to improve mechanistic understanding. The interpretation of genotypic concordance is comparable to that of Mendelian randomization across spouses with two important distinctions. First, genotypic concordance will not capture partner interaction effects as germline DNA is fixed for both spouses prior to assortment. Second, concordance induced by assortment will be further attenuated compared to a Mendelian randomization approach."

Supplementary

“Assuming a population of unrelated individuals (1000 males M and 1000 females F) where a phenotype P is assorted on (individuals more phenotypically similar for P are more likely to pair-up).

If $P \sim N(0,1)$ and is influenced by genetic factors $G \sim N(0,1)$ such that:

$$\text{Cov}(G, P) = \begin{pmatrix} 1 & \sqrt{h^2} \\ \sqrt{h^2} & 1 \end{pmatrix} \text{ where } h^2 \text{ is the proportion of variation in } P \text{ explained by } G$$

It then follows that with assortment on P such that in each male female pair $\text{Cor}(M_P, F_P) = C$, that the expected Mendelian randomization and genetic correlation estimates capturing spousal assortment are equivalent to:

$$\begin{aligned} \text{Cor}(M_P, F_G) &= \text{Cor}(M_G, F_P) = \frac{\text{Cov}(M_G, F_P)}{\sigma_{M_G} \sigma_{F_P}} = \text{Cov}(M_G, F_P) = \begin{pmatrix} 1 & \sqrt{C} \sqrt{h^2} \\ \sqrt{C} \sqrt{1-h^2} & 1 \end{pmatrix} \\ &= C * \sqrt{h^2} * \sqrt{1-h^2} \end{aligned}$$

$$\begin{aligned} \text{Cor}(M_G, F_G) &= \frac{\text{Cov}(M_G, F_G)}{\sigma_{M_G} \sigma_{F_G}} = \text{Cov}(M_G, F_G) = \begin{pmatrix} 1 & \sqrt{C} \sqrt{h^2} \\ \sqrt{C} \sqrt{h^2} & 1 \end{pmatrix} \\ &= C * h^2 \end{aligned}$$

Comment 1.3

3. Possibility of horizontal pleiotropy

The rs1229984 genotype has been shown to be associated with alcohol use, previously and in the current study. But what if rs1229984 is also associated with for example drug use or nicotine dependence? Any of these, or other associated phenotypes, could in theory be a driving force in the mate choice (or mate retaining) process. Admittedly, the authors states in the limitations (line 437-439) that ‘Assortment independent of alcohol use cannot be completely ruled out...’. I believe, however, the paper would benefit from some more attention to this issue.

If the effect of rs1229984-similarity is about alcohol use we would expect to see no remaining rs1229984-correlation across spouses, after controlling for phenotypic alcohol behavior. In contrast, if there is still a spousal rs1229984-correlation after controlling for alcohol behavior, there would be reason to look for other associated phenotypes that mediate the effect. I believe such analyses could easily be performed and would add to the validity of findings.

Response

The possibility of horizontal pleiotropy is an important limitation in conventional Mendelian randomization analyses, where genetic variants can potentially influence phenotypes via multiple distinct pathways. However, we would argue that in the context of assortative mating, pleiotropy is less likely to be problematic. Although, variants may influence other phenotypes in the index individual, the mechanisms through which an individual's genotype could plausibly affect their spouse's phenotypes are likely to be social rather than biological. Assuming that rs1229984 also influences a non-social trait (e.g. Vitamin D), it is unlikely that Vitamin D levels in an individual would affect the alcohol consumption of their spouse. As the reviewer suggested above, drug use or cigarette smoking could potentially influence spousal phenotypes, but variation in an alcohol dehydrogenase gene influencing these behaviours independent of alcohol behaviour again seems unlikely. Furthermore, looking up rs1229984 in the GeneAtlas (Canela-Xandri, Oriol, Konrad Rawlik, and Albert Tenesa. "An atlas of genetic associations in UK Biobank." *Nature Genetics* 50.11 (2018): 1593) (see: <http://geneatlas.roslin.ed.ac.uk/phewas/?variant=rs1229984&representation=table>) there is very weak evidence that the variant is associated with two smoking behaviour variables (Current tobacco smoking; P=0.43 & smoking Status; P=0.42). Similarly, a phenome-wide scan in UK Biobank showed no strong evidence for obvious pleiotropic effects independent from alcohol consumption (<http://geneatlas.roslin.ed.ac.uk/phewas/?variant=rs1229984&representation=plot>). We are happy to include this sensitivity analysis in the manuscript at the editor's discretion.

In our Mendelian randomization analysis of spousal height, we used various sensitivity analyses including the I^2 statistic and MR-Egger to evaluate consistency of the effect estimate. Although, we were unable to apply these sensitivity analyses to alcohol consumption, our results for height suggest that, in the context of spousal assortment, substantial bias from horizontal pleiotropy is unlikely. Furthermore, given the similarities between the effect estimates from the observational and Mendelian randomization analyses, it is relatively implausible that the effect could be completely independent of alcohol behaviour. In this instance, the true causal phenotype would have to be almost perfectly correlated with alcohol consumption.

Measurement error on alcohol consumption will mean that adjusting for measured alcohol consumption is unlikely to fully attenuate the phenotypic association. If poorly measured alcohol use is adjusted for and a genetic effect remains, this could be because of horizontal pleiotropy or because of a residual effect of alcohol. The problems of adjusting for mediators as a way of investigating possible independent effects of an underlying risk factor has been discussed in detail previously (for example, in Richmond RC, et al. Challenges and novel approaches for investigating molecular medication. *Human Molecular*

Genetics 2016;25: R149-156). We are happy to add a discussion of this in the manuscript at the editor's discretion. To explore the possibility of confounding, we have instead adjusted the alcohol observational analyses for plausible confounders that could explain the results (height, education and recruitment centre), with results not deviating greatly.

Previous estimate: 0.38 (95% C.I. 0.37, 0.38)

Updated estimate: 0.37 (95% C.I. 0.36, 0.38)

Comment 1.4

4. Minor issues

i) It would be helpful with some more descriptive statistics, e.g. the distribution of the drinking variables.

Response

We have added descriptive statistics of the drinking variables to the supplementary material in Supplementary Table 1.

ii) The analyzed sample consists of (assumed) spouses, but same-sex spouses are excluded. It would be helpful to provide arguments for this exclusion.

Response

We have added some information on why possible same-sex spouses were excluded, relating to the method used to derive the spouse pairs. However, we are planning to send a request to UK Biobank to access the data. Dependent on the timeline/feasibility of accessing this data we are happy to incorporate sexual orientation data into this manuscript.

Methods

We excluded 4,866 potential couples who were the same sex (9.3% of the sample) as unconfirmed same sex pairs may be more likely to be false positives. Although sexual orientation data was collected in UK Biobank, access is restricted for privacy/ethical reasons and the variable is not included in the data showcase.

iii) Alcohol behavior is reported to be highly heritable (line 69) but no references are included. It would be good to cite e.g. some twin studies and to include heritability estimates.

Response

We have added references and an estimate of the heritability of alcohol use disorders and alcohol consumption.

Methods

Alcohol behaviour has been shown to be highly heritable with estimates of 30-50% for alcohol use disorders (19,20) and a common variant heritability estimate of 13% for self-reported alcohol consumption (21).

19. Verhulst B, Neale MC, Kendler KS. The heritability of alcohol use disorders: a meta-analysis of twin and adoption studies. *Psychological Medicine* 2015;45(5):1061-72.
20. Walters GD. The heritability of alcohol abuse and dependence: a meta-analysis of behavior genetic research. *The American Journal of Drug and Alcohol Abuse* 2002;28(3):557-84.
21. Clarke T-K, Adams MJ, Davies G, et al. Genome-wide association study of alcohol consumption and genetic overlap with other health-related traits in UK Biobank (N= 112 117). *Molecular Psychiatry* 2017;22(10):1376.

iv) Analyzing height as a way of validating the subsequent alcohol analyses represents a strength. However, it is not fully clear why height is subjected to phenotypic correlation analyses and MR, but not genotypic correlations across spouses.

Response

We have now added this analysis into the manuscript, replicating previous findings of spousal genotypic concordance for height related loci (see: Robinson, Matthew R., et al. "Genetic evidence of assortative mating in humans." *Nature Human Behaviour* 1.1 (2017): 0016 and Yengo, Loic, et al. "Imprint of assortative mating on the human genome." *Nature Human Behaviour* 2.12 (2018): 948.).

v) The authors seem to use the term 'correlation' when referring to an association (e.g. "We estimated the spousal correlation ... using a logistic regression", line 179-181). This might be fine, but many readers would expect a correlation to mean e.g. 'Pearson correlation' or 'polychoric correlation'. It might be helpful to a) either refer to associations as 'associations' rather than 'correlations', or b) use 'correlations' but then also include estimates of Pearson and tetrachoric correlations (which would make the findings even more comparable to previous studies reporting such correlations).

Response

We have gone through the manuscript and changed language to be more consistent; correlation has been reworded to concordance although given that our models were scaled so that the exposure and outcome had identical distributions, we believe that our reported associations are largely equivalent to correlation estimates.

Reviewer 2

Comment 2.1

I notice that they initially frame the paper with a public health angle, including in the first paragraph of the introduction. I think that this is possibly less relevant - I don't think that the finding that spouses have similar health-related behaviors is particularly novel, and the public health relevance is not strengthened by demonstrating this with genetic instruments.

One thing that might be of public health interest is the balance of the direction of the effect. If high consuming partners increase consumption in their spouses or if tee-total people decrease the effect in theirs. One way to look at this might be to specifically consider the patterns of alcohol consumption within couples discordant for the variant studied here. This might suggest that you get a double benefit from treating one member of a couple, or that the people might slip back into old habits if both aren't considered.

Response

We agree that the relevance of our findings to public health was probably overstated and have therefore tweaked the introduction to downplay this assertion. We have instead stated that the mechanism is relevant from a biosocial perspective.

“The mechanism explaining spousal concordance for alcohol consumption could have important implications relating to human social and reproductive behaviour.”

In response to the second paragraph: although there are limitations of the analyses, we believe that the analyses aiming to explore relationship length and phenotypic similarities (in response to 1.1) do not provide strong evidence for partner interaction effects.

Comment 2.2

In terms of the analysis presented, there are a few things I would suggest. Firstly there seems to me that there is an risk in this analysis of the result being driven by assortative mating other than that based on alcohol consumption, a point that the authors acknowledge in the discussion. This is also suggested by the fact that the variant in question was associated with both some of the principle components and with birth site. What does the association look like when the values for one or both partners for these principle components are included in the model?

Response

We agree that it is important to comprehensively explore the possibility of bias from population stratification. As the reviewer suggested, we have sequentially added principal components to the Mendelian randomization analysis and found that including principal components in the model reduces estimate of the effect of an individual's SNP on their own alcohol consumption and their partner's alcohol consumption with the effect estimate relatively stable after the fourth principal component has been added. We therefore reran the Mendelian randomization analyses including the first 10 principal components of the reference individual and their partner (where relevant). In response to the third reviewer's

comments, we also included age, partner's age, centre and chip. Despite the changes in the table below, with both estimates in the table below decreasing, the Wald ratio estimator (causal effect estimate) remained largely consistent, although slightly attenuated.

Previous: 0.29 (95% C.I. 0.20, 0.38; $P=2.15 \times 10^{-9}$)

Updated: 0.26 (95% C.I. 0.15, 0.38; $P=1.10 \times 10^{-5}$)

Sequential addition of principal components to the SNP-Alcohol model

Principal components	rs1229984 and own alcohol consumption	rs1229984 and partner's alcohol consumption ¹
	Alcohol units per each additional major allele (95% C.I.)	Alcohol units per each additional major allele (95% C.I.)
Baseline model: Sex+Age	4.8 (4.3, 5.2)	1.4 (1.0, 1.9)
+PC1	4.4 (4.0, 4.9)	1.1 (0.7, 1.6)
+PC2	4.4 (3.9, 4.8)	1.1 (0.7, 1.6)
+PC3	4.2 (3.7, 4.6)	1.1 (0.6, 1.5)
+PC4	3.9 (3.5, 4.4)	1.0 (0.6, 1.5)
+PC5	3.9 (3.5, 4.4)	1.0 (0.6, 1.5)
+PC6	3.9 (3.5, 4.4)	1.0 (0.6, 1.5)
+PC7	4.0 (3.5, 4.4)	1.0 (0.6, 1.5)
+PC8	4.0 (3.5, 4.4)	1.0 (0.6, 1.5)
+PC9	3.9 (3.5, 4.4)	1.0 (0.6, 1.5)
+PC10	4.0 (3.5, 4.4)	1.0 (0.6, 1.5)

¹ Adjusted for reference individual and spouse PCs

Comment 2.3

For the analysis presented in Table 2, I notice that there seems to be a large difference in the numbers of spouse pairs included in this analysis from different regions. I suspect that this reflects different rates of migration to different cities (resulting in couples from nearer or further apparent), to what extent do the authors feel that this suggests limitations to the generalisability of the results? This locus is known to be ancestry specific, as are the social factors that lead to assortative mating. And, this may even be different in different parts of the UK. While there isn't any evidence of heterogeneity in the analysis, this might simply be due to a lack of diversity in the samples available in this analysis.

Response

Indeed, migration and contextual social differences complicate the interpretation of our findings. We initially attempted to disentangle some of these issues using the within-centre analysis and now, after reviewer feedback, have included further analyses to explore

potential issues (see responses to the third reviewer). Furthermore, the inclusion of the first 10 PCs in response to 2.2, did not greatly affect our findings.

In the discussion, we had previously discussed potential contextual influences that may mean our findings cannot be extrapolated to East Asian populations. We agree that there may also be regional differences across the UK and so have incorporated this caveat into the discussion.

“Indeed, even within the UK, there may be regional variation that we were unable to detect in this study.”

Comment 2.4

I'm curious to know if UK Biobank had a policy of people bringing their spouses along - was there any ad-hoc enrollment of this sort? I'm not sure that this would cause an issue with the analysis, but I can see that it might result in a sample where spouses were more similar on a range of factors than the general population.

Response

The issue of selection into UK Biobank, particularly with regards to spouses enrolling together is a concern. We contacted UK Biobank with this query and they responded as follows “We invited people at the same address so that could share travel etc. so quite often people/spouses did come together for that reason”. We have added spouse participation as a potential limitation in the discussion.

“Fourth, selection into the UK Biobank, particularly with regards to participation of spouse-pairs is a potential source of bias”.

Comment 2.5

The statistical analysis excluded those more than 5SDs. Why was it done? Do the authors think that this data is less reliable? If there interest was in public health, these potentially alcohol abusers might be a particular group to target.

Response

We decided to remove a small number of pairs that included extreme outliers (N=189), (e.g. individuals self-reporting a weekly alcohol consumption of over 120 units), for two reasons. First, as the reviewer suggested, this extreme data may be less reliable and more prone to measurement error. Second, from a statistical perspective, such extreme outliers may distort the analyses. In response to a comment from the first reviewer, we have added distributions of the alcohol variables, including a histogram to the supplementary material.

Reviewer 3

Comment 3.1

1) The authors observe that the SNP rs1229984 is significantly associated with the first Principal Components (PC), and birth coordinates (BC). As the authors already comment, this may be due to a population structure (which may be both local and/or global). Furthermore, this SNP may be pleiotropic. This could considerably affect the interpretation of the results. The authors address this point by performing analyses grouping individuals by assessment center. However, I think stronger evidence is needed to study how this stratification may affect the results. Below I make some suggestions which, in my opinion, would help to address this problem:

1.a) The authors showed that the rs1229984 genotypes are correlated with PC/BC. I think it would be useful to know if alcohol consumption is also associated with PC/BC.

Response

We have added relevant analyses to determine if self-reported alcohol consumption is also associated with principal components and birth coordinates. We find strong evidence that alcohol consumption is associated with these variables, particularly with the North-South axis, with the direction of effects regarding alcohol consumption concordant with the geographical stratification on rs1229984.

Supplementary Table 4: Association of self-reported weekly alcohol consumption with genetic principal components and birth coordinates

Principal components	385,287 individuals of European descent		337,114 individuals of White British descent	
	Beta (95% C.I.) ¹	P-value	Beta (95% C.I.) ¹	P-value
PC1	<10 ⁻¹⁶		3.3x10 ⁻¹¹	
PC2	<10 ⁻¹⁶		0.57	
PC3	<10 ⁻¹⁶		3.2x10 ⁻⁶	
PC4	<10 ⁻¹⁶		<10 ⁻¹⁶	
PC5	<10 ⁻¹⁶		<10 ⁻¹⁶	
PC6	<10 ⁻¹⁶		0.36	
PC7	0.18		1.1x10 ⁻⁵	
PC8	2.2x10 ⁻¹⁰		0.0094	
PC9	<10 ⁻¹⁶		2.9x10 ⁻¹⁰	
PC10	<10 ⁻¹⁶		1.8x10 ⁻¹²	
Birth-coordinates (units)	Beta (95% C.I.)¹	P-value	Beta (95% C.I.)¹	P-value
North-South Axis (kilometres north)	0.19 (0.16, 0.22)	<10 ⁻¹⁶	0.18 (0.15, 0.21)	<10 ⁻¹⁶
East-West Axis	-0.03 (-0.04, -	7.1x10 ⁻⁵	-0.03 (-0.04, -	4.4x10 ⁻⁴

(kilometres east)	0.02)		0.01)	
-------------------	-------	--	-------	--

1 Per 1 unit increase in weekly alcohol consumption

1.b) I think that additional variables have to be included in the models that estimate the SNP effect. In addition to sex, the first principal components, age (I think that age, and partner age may also be relevant when computing correlation of alcohol consumption between partners), genotyping array and assessment center should also be included. In addition, given the stratification of the SNP, and that there are relatives in the analysis, I think that the ordinary linear models (i.e. the non-logistic models used for estimating the SNP effect), would benefit from using Mixed Linear Models to correct by the Genetic Relationship Matrix (using a schema such as leave-one-chromosome-out, LOCO, where the chromosome of the tested SNP is excluded from the GRM, otherwise they may lose power. Tools such as BOLT-LMM will do this automatically).

Response

In response to the reviewer's suggestions, we have added age and partner's age to the observational analysis, and age, partner's age, principal components (of both spouses where relevant) and centre into the Mendelian randomization analysis. We are reluctant to add genotyping chip as a covariate into the model, as this may induce collider bias with one of the genotyping arrays largely sampling smokers. We believe the addition of these covariates is sufficient, with similar additions proposed by two of the reviewers, but are happy to also use a more complex model (e.g. BOLT-LMM) at the editor's discretion.

1.c) Given that rs1229984 is stratified, why the authors did not also used other alcohol consumption associated SNPs in the same way the authors do with different SNPs associated with height?

Response

We initially considered including other SNPs associated with alcohol consumption in analyses. However, after some discussion we decided to restrict analyses to the *ADH1B* variant for two reasons. First, rs1229984 is a well-characterised variant with a considerable effect on alcohol consumption that has been used extensively in Mendelian randomization analyses. Second, beyond rs1229984, there are few strong candidates for alcohol instruments in European populations. Previous studies have replicated a few other loci, but have limitations such as; 1) modest effect sizes in comparison to rs1229984, 2) the SNPs were identified in the same population as this study (UK Biobank) and 3) adjusted for BMI, which could be problematic in a Mendelian randomization framework (see: Holmes, M.V. and Smith, G.D., 2018. Problems in interpreting and using GWAS of conditional phenotypes illustrated by 'alcohol GWAS'. *Molecular Psychiatry*, p.1.). For these reasons, using rs1229984 is the best option available at present in the context of this study.

1.d) The authors address the problem of the SNP stratification by performing analyses by

assessment center. I think these analyses provide some support for the observed results not being artifacts from stratification. However, I think it would also be good to show that the genotype of the individuals grouped by assessment center, are not associated with their PC/BC. In addition, I would use a somewhat inverse approach. Are the genotypes of the couples that were born 100 or 200 (a number as large as possible, but that still retains a considerable sample size) miles away still correlated? If so, is their alcohol consumption also correlated? If the answers to this question is yes, then: what is the effect of rs1229984 on the partner alcohol consumption for this couple?

Response

We have included extra analyses exploring if spousal differences in birth coordinates and principal components are associated with spousal differences in alcohol consumption/rs1229984 genotype.

We find that both birth coordinates and principal components are strongly associated with alcohol consumption and rs1229984 genotype in the complete spousal comparison analysis. When restricting to spouses only born within 100 km (as in the sensitivity analysis), there is no strong evidence for an association between birth coordinates and self-reported alcohol consumption or rs122994 genotype, with a major caveat that measurement error of birth coordinates, which are rounded to the nearest km, may bias these results towards the null. Differences in genomic principal components remained associated with alcohol consumption and rs122984 genotype differences in the within-centre analysis emphasising the fine-scale population structure of the UK Biobank. See Supplementary Table 5.

Comment 3.2

2) Related to the model used for estimating the effect of the SNP rs1229984 on alcohol consumption, the rationale behind using a dominant model vs an additive model is not clear to me. Although the MAF is small, I think that with this sample size, there will be enough individuals with two minor alleles to fit the models. Although, there will be a small number of individuals (~100?) with two minor alleles, the effect of the minor/major allele may be under/over estimated. Maybe it is not relevant, but without checking, it is hard to test their potential effect. Given that the SNP effects estimated in the model may be relevant for the Mendelian Randomization approach, and given that some confidence intervals are overlapping by a small margin, I think it would be relevant to at least test whether the estimated SNP effects on alcohol consumption are affected by fitting an additive model.

Response

We agree that the large sample size of UK Biobank should be sufficient to explore the most appropriate model for the low frequency rs1229984 SNP. Therefore, we evaluated

the median self-reported alcohol consumption for each genotype in the European and White British samples.

Indeed, the results support an additive effect of rs1229984 on alcohol consumption. All analyses have been rerun using an additive model but given the modest number of minor allele homozygotes, had modest effects and did not affect our conclusions. Nevertheless, it is useful for future work to know that an additive model is most appropriate for the variant.

Supplementary Table 1: Weekly alcohol consumption per rs1229984 genotype

	385,287 individuals of European descent			337,114 individuals of White British descent		
	CC (N=363,036) Median (Q1, Q3)	TC (N=20,146) Median (Q1, Q3)	TT (N=569) Median (Q1, Q3)	CC (N=320,699) Median (Q1, Q3)	TC (N=14,728) Median (Q1, Q3)	TT (N=188) Median (Q1, Q3)
Weekly alcohol consumption (units)	12.0 (0, 24)	7.0 (0, 17.5)	2.0 (0, 13)	12.0 (0, 24.0)	8.0 (0, 18)	4.3 (0, 18.1)

Comment 3.3

Minor points:

- In figure 1, would it make sense to add an additional possibility after spousal pairing? A diagram were a third factor "X" affecting alcohol consumption of both partners (a factor which may also be correlated with genetic variants)?.

Response

We believe that the suggested addition by the reviewer is similar to the social homogamy point in that they are both possibilities that could be termed as "confounding". In response to the first reviewer, we added relationship dissolution as a further possibility and are reluctant to complicate the figure further.

- In figure 1, the legend is labeled A), B), and C), but the figure as 1), 2), and 3).

Response

We have ensured that figure labels match the legend labels.

- The authors comment about rs1229984 violating the HWE hypothesis. As a suggestion, I would also add that HWE may be violated under selection (the authors point out that this SNP may be under selection in the Discussion), In addition, I think the HWE threshold should be much less restrictive when the sample sizes start to be large enough (and the population structure potentially increases its effect on the test). For instance, on the PLINK2.0 website they comment about this (and they suggest to use a small threshold for filtering, <https://www.cog-genomics.org/plink/2.0/filter#hwe>).

Response

As the reviewer stated, selection could also lead to violation of HWE. However, given that the SNP violated HWE in a European population and this did not seem to be the case in a homogeneous White British subset, it seems more likely that population substructure differences explain our findings. For completeness, we have added the following:

“Of further interest is that the variant has previously been shown to be under selection⁶¹ suggesting that the variant has historically had a substantial effect on reproductive fitness **and may partially explain the violation of HWE observed across Europeans in our analyses.**”

Concerning the comment on the HWE threshold, we would argue that a threshold is more relevant to quality control on genotyping quality. In this study, we were more interested in HWE with respect to violation of HWE being a consequence of assortative mating.

- The authors use several variables to decide whether two individuals are part of a couple. However, they do not use shared household information which may be available under request. If it is not too much hassle, it would be interesting to see whether the computed couples agree with the shared household information.

Response

We contacted UK Biobank regarding this comment and they replied as follows:

“There have been some attempts in the past to generate “shared household” data, both internally and then through an external project. However, some questions were raised about how successful the methodologies of these approaches had been, and so this project was put on hold and is not currently actively moving forward. As a result we do not have any such information that we could easily make available at this time.”

- It would be easier for the reader if the author indicate the UK Biobank codes of the used fields, when these are first described in the methods.

Response

We have added relevant UK Biobank field codes to the methods section.

- I think there is a typo in the line 192: an -> a.

Response

We have updated this.

- In line 312, I believe it would be clearer if it says "...than the reference group.", "...the partners of the reference group."

Response

We have amended this sentence as suggested.

- Overall the manuscript is clearly written and easy to understand. However, I had sometimes the feeling that the manuscript is a bit disordered. Maybe putting the sections about height in the Results section together would help. If the Methods will be in the main text, there are some bits that also seem a bit redundant. However, this is just my opinion, which may be based on a personal preference, so feel free to ignore it if the authors disagree.

Response

We have reordered the manuscript as suggested, putting the height sections together, which will hopefully improve the clarity and flow of the piece.

Reviewers' Comments:

Reviewer #1:

Remarks to the Author:

The authors have done a solid job in revising the paper. They have addressed the issues raised in the initial review, and have revised the manuscript thoroughly. In general, I believe the revision and response letter concerning my comments are satisfactory. I have however a few minor issues that might be worth considering:

1. Terminology.

The authors use "genetic risk scores (GRS)" to denote the polygenic scores for height. Although not uncommon I believe the term 'risk' is somewhat misleading for height (and other non-disorder phenotypes) and would suggest the term "polygenic scores".

2. Mendelian randomization across spouses.

The use of MR for the same phenotype in two individuals is important and a substantial strength of the current study. As I am not an expert in this design I have a somewhat naive question (which other non-experts also might wonder about).

The MR analysis estimated that a 1 unit increase in an individual's alcohol consumption increased their partner's use by 0.26 units (cf. abstract). Given a reciprocal relationship, alcohol use in spouse 1 will increase alcohol use in spouse 2, and correspondingly, alcohol use in spouse 2 will increase alcohol use in spouse 1. Thus, the effect from spouse 1 to spouse 2 will assumedly only represent half the total correlation between spouses, or in other words, the observed correlation should be twice the effect from one spouse to the other. How are these two effects separated in the analyses?

I assume that the dual process involved, with genotype1->phenotype1->phenotype2 and similarly genotype2->phenotype2->phenotype1 both contributing to the observed associations, is incorporated into the analytic methods and the equations in Supplementary Methods. However, it might be helpful to many readers to provide a brief outline of this issue, and a few explanatory comments.

Reviewer #2:

Remarks to the Author:

I'm very pleased with the authors response to the reviews, and have no further comments on this manuscript.

Thank you.

Reviewer #3:

Remarks to the Author:

This new version of the manuscript is improved and the authors have done a great job answering to many of my comments. However, I believe some of the main points I raised in the first review still require some further analysis. I have still some concerns about the population structure. Some of the new analyses indicate that both alcohol consumption and the SNP used as instrumental variable are correlated with the population structure. Furthermore, some of these new analyses also suggest that this may not only happen globally, but also in a more local manner (i.e. when stratifying by assessment center). The interpretation of the results (e.g. that assortative mating is creating a correlation between the couples' SNP variants), may be affected by this. I think that it would be relevant to try to further assess the importance of this structure, which I think it is still unclear.

1. Given that the authors indicate they cannot use other SNPs as instrumental variables for alcohol consumption, could the authors use the couples born 100 miles away (or even 200 miles, or both) to estimate the instrumental SNP's effect on the alcohol consumption of both the individuals and their partners? Could they use these couples to estimate the effect of an individual's alcohol consumption on their partner's alcohol consumption? Could they also compute the genotypic concordance of of this SNP between these couples? In these analyses the available number of couples will be smaller, and the statistical power may not be enough. But it would be useful to check if the estimated effects and confidence intervals in these couples born far away (hopefully diluting considerably the population structure) are in agreement with the ones already estimated. If the effect sizes are different, they may still be relevant. But the interpretation of the results may change (although the authors already did a great job discussing different possibilities about the interpretation of the results in the Discussion section).

Minor points:

1. The authors already used these couples born 100 miles away in Supplementary Table 6 to estimate some associations. It is not clear to me what they exactly tested. When they say "genotype differences", do they mean the differences on the number of ref./alt. allele copies between the couples? I am not sure about how the authors interpret these results.

2. There seems to be a typo in lines 408, and 412. If I understand properly, in line 412 the authors indicate that 47,321 couples remain after removing outliers from 44,886 couples. Maybe I am misinterpreting what they are describing, but it does not sound right that there are more couples after filtering.

Reviewer #1 (Remarks to the Author):

The authors have done a solid job in revising the paper. They have addressed the issues raised in the initial review and have revised the manuscript thoroughly. In general, I believe the revision and response letter concerning my comments are satisfactory. I have however a few minor issues that might be worth considering:

1. Terminology.

The authors use “genetic risk scores (GRS)” to denote the polygenic scores for height. Although not uncommon I believe the term ‘risk’ is somewhat misleading for height (and other non-disorder phenotypes) and would suggest the term “polygenic scores”.

Author response:

We agree that “risk of height” is misleading; we have changed “genetic risk scores (GRS)” to “polygenic scores (PGS)” throughout the manuscript.

2. Mendelian randomization across spouses.

The use of MR for the same phenotype in two individuals is important and a substantial strength of the current study. As I am not an expert in this design I have a somewhat naive question (which other non-experts also might wonder about).

The MR analysis estimated that a 1 unit increase in an individual’s alcohol consumption increased their partner’s use by 0.26 units (cf. abstract). Given a reciprocal relationship, alcohol use in spouse 1 will increase alcohol use in spouse 2, and correspondingly, alcohol use in spouse 2 will increase alcohol use in spouse 1. Thus, the effect from spouse 1 to spouse 2 will assumedly only represent half the total correlation between spouses, or in other words, the observed correlation should be twice the effect from one spouse to the other. How are these two effects separated in the analyses?

I assume that the dual process involved, with genotype1->phenotype1->phenotype2 and similarly genotype2->phenotype2->phenotype1 both contributing to the observed associations, is incorporated into the analytic methods and the equations in Supplementary Methods. However, it might be helpful to many readers to provide a brief outline of this issue, and a few explanatory comments.

Author response:

After further discussion we have devised directed acyclic graphs (DAGs) for the phenotypic concordance, Mendelian randomization and genotypic concordance analyses. We believe that these DAGs will better illustrate the mechanisms by which

an individual's alcohol behaviour can affect the alcohol behaviour of their spouse. We have decided to replace the old Figure 2 (see below) with these DAGs (see the next page) as we believe that the updated figure largely supersedes the older figure.

With respect to the interpretation of Mendelian randomization analysis, the MR analysis estimate is likely to be smaller than an unbiased phenotypic estimate for two reasons. First, because assortment is on phenotype, the genotype-phenotype correlation is expected to be smaller (demonstrated in the supplementary material). Second, as the reviewer has noted, the Mendelian randomization estimate will capture only partner interaction effects of the index individual on their partner and not partner interaction effects of the partner on the index individual (as genotype is fixed). We have added detail on this to the main text.

“Note that as genotype is fixed from birth, a Mendelian randomization estimate will not capture an effect of the partner’s alcohol consumption on the index individual during the relationship.”

Old Figure 2

New Figure 2 and descriptions

(A) Phenotypic concordance:
 $A_I^R \sim A_P^R$

(B) Mendelian randomization:
 $Z_I \sim A_P^R$

(C) Genotypic concordance:
 $Z_I \sim Z_P$

Key

$Z_{I,P}$: *ADH1B* genotype for an individual (I) and their partner (P).

$A_{I,P}^O$: alcohol consumption prior to partnering for an individual (I) and their partner (P).

$A_{I,P}^R$: alcohol consumption in the relationship for an individual (I) and their partner (P).

C a measure of mate choice.

D a measure of relationship duration.

U representing unmeasured confounding factors.

Green: Effects prior to relationship formation.

Blue: Effects while spouses are in a relationship.

“Figure 2 Directed acyclic graphs illustrating phenotypic concordance, Mendelian randomization and genotypic concordance analyses between-spouses.

(A) Phenotypic concordance. Spousal concordance for alcohol use during their relationship (A_I^R, A_P^R), as measured in UK Biobank, could be explained by several different possibilities.

Assortative mating: Alcohol consumption prior to relationship formation (A_I^O, A_P^O) influences mate choice C which induces spousal correlations for A^O and A^R .

Partner interaction effects: An individual’s alcohol consumption behaviour at relationship formation (A_I^O, A_P^O) influences spousal alcohol behaviour during the relationship (A_I^R, A_P^R). Note that effects likely relate to relationship length.

Relationship dissolution: Spousal alcohol behaviour during the relationship (A_I^R and A_P^R) influences the duration of the relationship D , which in turn induces spousal correlations for A^R in the remaining couples.

Confounding factors: Unmeasured confounders U influence both C and A^O leading to spousal correlation for A^R independent of an effect of A^O on C .

(B) Mendelian randomization framework. An association between an individual’s alcohol influencing genotype Z_I and their spouse’s alcohol use A_P^R suggests that the spousal concordance is explained by **assortative mating, partner interaction effects** or **relationship dissolution**. Genetic variants are unlikely to be associated with socio-economic confounders suggesting that the **confounding factors** possibility is unlikely.

(C) Genotypic concordance. Genotypic concordance for alcohol related genetic variants (Z_I, Z_P) suggests that some degree of the spousal concordance is explained by **assortative mating** or **relationship dissolution**. **Partner interaction effects** cannot lead to genotypic concordance because genotypes are fixed from birth.

Reviewer #2 (Remarks to the Author):

I'm very pleased with the authors response to the reviews, and have no further comments on this manuscript.

Thank you.

Author response:

We thank the reviewer for their time and positive feedback.

Reviewer #3 (Remarks to the Author):

This new version of the manuscript is improved and the authors have done a great job answering many of my comments. However, I believe some of the main points I raised in the first review still require some further analysis. I have still some concerns about the population structure. Some of the new analyses indicate that both alcohol consumption and the SNP used as instrumental variable are correlated with the population structure. Furthermore, some of these new analyses also suggest that this may not only happen globally, but also in a more local manner (i.e. when stratifying by assessment center). The interpretation of the results (e.g. that assortative mating is creating a correlation between the couples' SNP variants), may be affected by this. I think that it would be relevant to try to further assess the importance of this structure, which I think it is still unclear.

1. Given that the authors indicate they cannot use other SNPs as instrumental variables for alcohol consumption, could the authors use the couples born 100 miles away (or even 200 miles, or both) to estimate the instrumental SNP's effect on the alcohol consumption of both the individuals and their partners? Could they use these couples to estimate the effect of an individual's alcohol consumption on their partner's alcohol consumption? Could they also compute the genotypic concordance of this SNP between these couples? In these analyses the available number of couples will be smaller, and the statistical power may not be enough. But it would be useful to check if the estimated effects and confidence intervals in these couples born far away (hopefully diluting considerably the population structure) are in agreement with the ones already estimated. If the effect sizes are different, they may still be relevant. But the interpretation of the results may change (although the authors already did a great job discussing different possibilities about the interpretation of the results in the Discussion section).

Author response:

As the reviewer suggested, we have run the Mendelian randomization and genotypic concordance analyses in the subsample of 13,770 pairs born more than 100 km

apart. Additionally, we have added the key results from these analyses to a supplementary table with several mentions in the main text.

Methods

“As a further sensitivity analysis, we compared Mendelian randomization and genotypic concordance estimates between the sample of 28,653 spouse-pairs born within 100 kilometres of each other with estimates from the sample of 13,770 pairs born more than 100 kilometres apart.”

Results

*“Furthermore, confidence intervals universally overlapped between Mendelian randomization and genotypic concordance estimates when comparing estimates from spouse-pair samples stratified on geographical birth proximity (born within 100km and born more than 100km apart) and the full spouse-pair sample (**Supplementary Table 5**).”*

Discussion

“Additionally, we conducted sensitivity analyses, including a within centre sensitivity analysis excluding spouse-pairs born more than 100 kilometres apart and a stratification of spouse-pairs by geographical birth proximity, finding consistent effect estimates.”

Results for reviewer (not contained in main text)

Mendelian randomization analysis

In this subsample, each alcohol increasing allele was associated with a 4.10 unit-increase (95% C.I. 3.27, 4.94) in that individual and a 0.79 unit-increase (95% C.I. -0.05, 1.63) in their partner. The Mendelian randomization estimate of the effect of an individual's alcohol consumption on their partner's was: 0.19 (95% C.I. -0.01, 0.40).

For comparison, in the sample of 28,653 pairs born less than 100 km apart, the corresponding estimates were 3.94 units a week (95% C.I. 3.39, 4.50), 1.16 units (95% C.I. 0.60, 1.75) and 0.29 (95% C.I. 0.15, 0.44) respectively.

For comparison, in the full sample, the corresponding estimates were 3.99 units a week (95% C.I. 3.52, 4.45), 1.06 units a week (95% C.I. 0.59, 1.52) and 0.26 (95% C.I. 0.15, 0.38) respectively.

While the causal effect estimate was slightly lower in the sample of spouse pairs born more than 100km apart, confidence intervals heavily overlap with the estimates from the full sample (-0.01, 0.40; 0.15, 0.38) limiting further conclusions.

Genotypic concordance analysis

The genotypic concordance estimate for 13,770 pairs born more than 100 km apart was 0.009 (95% C.I. -0.008, 0.026). For comparison, in the sample of pairs born less than 100 km apart and in the full sample, the estimates were 0.017 (0.005, 0.029) and 0.019 (95% C.I. 0.010, 0.028) respectively. Similarly, while the effect estimate is slightly lower in magnitude in the sample of spouse pairs born more than 100km apart, confidence intervals overlap preventing stronger conclusions.

Minor points:

1. The authors already used these couples born 100 miles away in Supplementary Table 6 to estimate some associations. It is not clear to me what they exactly tested. When they say "genotype differences", do they mean the differences on the number of ref./alt. allele copies between the couples? I am not sure about how the authors interpret these results.

Author response:

In Supplementary Table 4 (table numbers modified with methods sections being moved), "Spousal rs1229984 genotype differences" refers to differences in the number of reference/alternative alleles between the two members of each couple. We used these analyses to evaluate whether restricting to spouse-pairs born within 100km of each other reduces the effects of possible population stratification bias. Our interpretation is that restricting to couples born closer together may reduce bias as birth location differences within this sample did not seem to be strongly associated with alcohol behaviour or rs1229984 differences. However, as noted in the manuscript, we did find some evidence that genomic principal component differences were associated with these alcohol drinking factors.

2. There seems to be a typo in lines 408, and 412. If I understand properly, in line 412 the authors indicate that 47,321 couples remain after removing outliers from 44,886 couples. Maybe I am misinterpreting what they are describing, but it does not sound right that there are more couples after filtering.

Author response:

We have corrected the sample sizes in this section of the manuscript.

"For self-reported alcohol consumption volume; 47,510 spouse-pairs had either complete phenotype data or reported their consumption frequency as less than weekly (in which case their weekly volume was assumed to be 0). After removing

189 pairs with outlying values (>5 S.D from the mean) from one or more members, the final sample included 47,321 spouse-pairs.”

Reviewers' Comments:

Reviewer #3:

Remarks to the Author:

I believe the authors did a great job addressing all my comments and I do not have further comments regarding my previous points. However, I still have a few minor points I believe should be addressed.

- I did not realise this previously (I apologise for this). The equations developed in Supplementary Methods (section "Assortment: theory and simulations") may have a problem with the variances. There, the authors define $P \sim N(0,1)$ and $G \sim N(0,1)$. However, I think this is not possible if there is environmental noise uncorrelated with G (which I believe is an assumption the authors make afterwards). For instance, let's assume the simplest model where $P = G + E$, $G \sim N(0,1)$, and $E \sim N(0,1)$. If we also assume that E and G are uncorrelated, then $P \sim N(0, 1 + 1)$. If I am not looking at this wrong, the variance of P cannot be 1 unless the variance of E is 0.

- I believe Figure 2 represents many possible interactions and it is (at least for me) a bit confusing. But given the detail of paths, I think the figure is missing some of them. For instance, why there are no arrows in both directions between Arp and Ari in panels (A) and (B)? In panel (B) Z_p is correlated with Z_i , and this may lead to a violation of MR assumptions (as the authors indicate in the Discussion). Shouldn't this be represented there?

- I finally have a minor comment regarding the labeling of Supplementary Tables. If I am not mistaken, it seems there are some Supp. Tables not properly labeled. For instance:

* line 189, should it be Supp. Table 4, instead of Supp. Table 5?

* line 215, should it be Supp. Table 5, instead of Supp. Table 7?

I did not check all tables, but there may be others. The authors should check the labeling. Or perhaps reorder the tables in the Supp. Material.

Reviewers' comments:

Reviewer #3 (Remarks to the Author):

I believe the authors did a great job addressing all my comments and I do not have further comments regarding my previous points. However, I still have a few minor points I believe should be addressed.

Response

We are pleased that the reviewer is happy with our previous response and appreciate the reviewer's thorough examination of our work.

Comment

- I did not realise this previously (I apologise for this). The equations developed in Supplementary Methods (section "Assortment: theory and simulations") may have a problem with the variances. There, the authors define $P \sim N(0,1)$ and $G \sim N(0,1)$. However, I think this is not possible if there is environmental noise uncorrelated with G (which I believe is an assumption the authors make afterwards). For instance, let's assume the simplest model where $P = G + E$, $G \sim N(0,1)$, and $E \sim N(0,1)$. If we also assume that E and G are uncorrelated, then $P \sim N(0, 1 + 1)$. If I am not looking at this wrong, the variance of P cannot be 1 unless the variance of E is 0.

Response

We thank the reviewer for noticing this, we have tweaked the notation in the supplementary material.

If $P \sim N(0, x)$ and is influenced by genetic factors $G \sim N(0, y)$ and non-genetic factors $E \sim N(0, x - y)$ such that:

The notation error did not influence our simulated data, but we have also modified "expected Mendelian randomization" in the text to "expected scaled Mendelian randomization" to reflect potential differences in variance between genotype and phenotype.

Comment

- I believe Figure 2 represents many possible interactions and it is (at least for me) a bit confusing. But given the detail of paths, I think the figure is missing some of them. For instance, why there are no arrows in both directions between Arp and Ari in panels (A)

and (B)? In panel (B) Z_p is correlated with Z_i , and this may lead to a violation of MR assumptions (as the authors indicate in the Discussion). Shouldn't this be represented there?

Response

In Figure 2, we have tried to graphically illustrate the possible influences on spousal similarities using a Directed Acyclic Graph (DAG). We appreciate the feedback on the figure, as it is challenging to condense complex relationships into a single figure. We have made some amendments to the figure and descriptions, detailed below. At the editor's discretion, we are also willing to provide further detail to these DAGs but we think such detail would be beyond the scope of this project.

In response to: "For instance, why there are no arrows in both directions between A_{rp} and A_{ri} in panels (A) and (B)?"

Spouses influencing each other over time is a complex stochastic process with time a continuous factor. Therefore, we attempted to simplify this graphically by including only two time points (O and R). The reviewer is correct in that an individual's alcohol use during the relationship can affect their spousal alcohol use. We have tried to represent this mechanism with the arrows from A_i^O to A_p^R , which occur during the relationship.

We decided against including arrows from A_i^R to A_p^R for two reasons. First, from a time perspective, it is previous alcohol use (e.g. at time R-1, R-2.. R-3...O) that is influencing alcohol use at point R. Furthermore, in a DAG, there cannot be cyclical relationships (e.g. arrows in both directions between A_i^R and A_p^R).

We have edited the Figure descriptions to better explain our model.

BEFORE:

Partner interaction effects: An individual's alcohol consumption behaviour at relationship formation (A_i^O , A_p^O) influences spousal alcohol behaviour during the relationship (A_i^R , A_p^R). Note that effects likely relate to relationship length.

AFTER:

Partner interaction effects: Spouses may influence each other's alcohol behaviour over time while in a relationship. We represent this stochastic process by the arrows between alcohol use at relationship formation (A_i^O , A_p^O) and alcohol use at study entry (A_i^R , A_p^R). Note that effects likely relate to relationship length.

In response to: "In panel (B) Z_p is correlated with Z_i , and this may lead to a violation of MR assumptions (as the authors indicate in the Discussion). Shouldn't this be represented there?"

The reviewer is correct that Z_I and Z_P may become correlated as a consequence of spousal assortment and this is of great interest. However, we would argue that because this is not a conventional causal effect (germline genotype cannot be modified), but rather a consequence of comparing assorted/non-dissolved spouses, it does not warrant a path between the two in the DAG. We have tweaked the figure descriptions to better illustrate this point.

BEFORE:

Assortative mating: Alcohol consumption prior to relationship formation (A_I^O, A_P^O) influences mate choice C which induces spousal correlations for A^O and A^R .

Relationship dissolution: Spousal alcohol behaviour during the relationship (A_I^R and A_P^R) influences the duration of the relationship D , which in turn induces spousal correlations for A^R in the remaining couples.

(C) Genotypic concordance. Genotypic concordance for alcohol related genetic variants (Z_I, Z_P) suggests that some degree of the spousal concordance is explained by **assortative mating** or **relationship dissolution**. **Partner interaction effects** cannot lead to genotypic concordance because genotypes are fixed from birth.

AFTER:

Assortative mating: Alcohol consumption prior to relationship formation (A_I^O, A_P^O) influences mate choice C . Comparing assorted pairs induces spousal correlations for A^O and A^R .

Relationship dissolution: Spousal alcohol behaviour during the relationship (A_I^R and A_P^R) influences the duration of the relationship D . Comparing non-dissolved pairs induces spousal correlations for A^R in the remaining couples.

(C) Genotypic concordance. Genotypic concordance for alcohol related genetic variants (Z_I, Z_P) suggests that some degree of the spousal concordance is explained by comparing assorted or non-dissolved pairs (**assortative mating/ relationship dissolution**). **Partner interaction effects** cannot lead to genotypic concordance because genotypes are fixed from birth.

Comment

- I finally have a minor comment regarding the labeling of Supplementary Tables. If I am not mistaken, it seems there are some Supp. Tables not properly labeled. For instance:

* line 189, should it be Supp. Table 4, instead of Supp. Table 5?

* line 215, should it be Supp. Table 5, instead of Supp. Table 7?

I did not check all tables, but there may be others. The authors should check the labeling. Or perhaps reorder the tables in the Supp. Material.

Response

We thank the reviewer for highlighting this, we have been through the manuscript and checked that the supplementary tables are correctly referenced.

Reviewers' Comments:

Reviewer #3:

Remarks to the Author:

I believe the authors did a great job addressing all my comments. I do not have any further comments.